# Near-Minimax-Optimal Distributional Reinforcement Learning with a Generative Model

**Mark Rowland**
Google DeepMind*

**Li Kevin Wenliang**
Google DeepMind

**Rémi Munos**
FAIR, Meta[†]

**Clare Lyle**
Google DeepMind

**Yunhao Tang**
Google DeepMind

**Will Dabney**
Google DeepMind

## Abstract

We propose a new algorithm for model-based distributional reinforcement learning (RL), and prove that it is minimax-optimal for approximating return distributions in the generative model regime (up to logarithmic factors), the first result of this kind for any distributional RL algorithm. Our analysis also provides new theoretical perspectives on categorical approaches to distributional RL, as well as introducing a new distributional Bellman equation, the stochastic categorical CDF Bellman equation, which we expect to be of independent interest. Finally, we provide an experimental study comparing a variety of model-based distributional RL algorithms, with several key takeaways for practitioners.

## 1 Introduction

In distributional reinforcement learning, the aim is to predict the full probability distribution of possible returns at each state, rather than just the mean return (Morimura et al., 2010a; Bellemare et al., 2017, 2023). Applications of distributional reinforcement learning range from dopamine response modelling in neuroscience (Dabney et al., 2020), to driving risk-sensitive decision-making and exploration in domains such as robotics (Bodnar et al., 2020), healthcare (Böck et al., 2022), and algorithm discovery (Fawzi et al., 2022), as well as forming a core component of many deep reinforcement learning architectures (Bellemare et al., 2017; Dabney et al., 2018b,a; Yang et al., 2019; Bellemare et al., 2020; Shahriari et al., 2022; Wurman et al., 2022).

The full distribution of returns is a much richer signal than the expectation to predict. A foundational, as-yet-unanswered problem is how many sampled transitions are required to accurately estimate return distributions, and in particular, whether this task is statistically harder than estimating just the value function. We study these questions in the setting where sampled transitions are given by a generative model (Kearns et al., 2002; Kakade, 2003; Azar et al., 2013).

We provide a new distributional RL algorithm, the *direct categorical fixed-point algorithm* (DCFP), and prove that the number of samples required by this algorithm for accurate return distribution estimation matches the lower bound established by Zhang et al. (2023), up to logarithmic factors. This resolves the foundational question above, and, perhaps surprisingly, shows that in this setting, *distributional RL is essentially no harder, statistically speaking, than learning a value function*.

In addition to this central result, our analysis provides new perspectives on categorical approaches to distributional RL (Bellemare et al., 2017), including a new distributional Bellman equation, the stochastic categorical CDF Bellman equation (see Section 5.2), which we expect to be of broad use in future work on categorical distributional RL. We also provide an empirical study, comparing

---

*Correspondence to `markrowland@google.com`. [†]Work done while at Google DeepMind.

38th Conference on Neural Information Processing Systems (NeurIPS 2024).

the newly-proposed DCFP algorithm to existing approaches to distributional RL such as quantile dynamic programming (QDP; Dabney et al., 2018b; Rowland et al., 2024), and identify several key findings for practitioners, including the importance of levels of environment stochasticity and discount factor for the relative performance of these algorithms.

## 2 Background

Throughout the paper, we consider the problem of evaluation in an infinite-horizon Markov reward process (MRP), with finite state space $\mathcal{X}$, transition probabilities $P \in \mathbb{R}^{\mathcal{X} \times \mathcal{X}}$, reward function $r : \mathcal{X} \to [0, 1]$, and discount factor $\gamma \in [0, 1)$; this encompasses the problem of policy evaluation in Markov decision processes (Sutton and Barto, 2018). A random trajectory $(X_t, R_t)_{t \geq 0}$ is generated from an initial state $X_0 = x$ according to the conditional distributions $X_t \mid (X_0, \ldots, X_{t-1}) \sim P(\cdot | X_{t-1})$, and $R_t = r(X_t)$. The *return* associated with the trajectory is given by the quantity $\sum_{t \geq 0} \gamma^t R_t$. In RL, a central task is to estimate the *value function* $V^* : \mathcal{X} \to \mathbb{R}$, defined by

$$V^*(x) = \mathbb{E}[\sum_{t \geq 0} \gamma^t R_t \mid X_0 = x], \tag{1}$$

given some form of observations from the MRP. The value function defines the expected return, conditional on each possible starting state in the MRP. The value function satisfies the Bellman equation $V^* = TV^*$, where $T : \mathbb{R}^{\mathcal{X}} \to \mathbb{R}^{\mathcal{X}}$ is defined by

$$(TV)(x) = \mathbb{E}_x[R + \gamma V(X')], \tag{2}$$

where $(x, R, X')$ is a random transition in the environment, distributed as described above. When the transition probabilities of the MRP are known, the right-hand side can be evaluated as an affine transformation of $V$. MRP theory (see, e.g., Puterman, 2014 for an overview) then shows how (an approximation to) $V^*$ can be obtained. For example, a *dynamic programming* approach takes an initial approximation $V_0 \in [0, (1-\gamma)^{-1}]^{\mathcal{X}}$, and the sequence $(V_k)_{k=0}^{\infty}$ is computed via the update $V_{k+1} = TV_k$; it is guaranteed that $\|V_k - V^*\|_\infty \leq \gamma^k (1-\gamma)^{-1}$. Alternatively, the linear system $V = TV$ can be solved directly with linear-algebraic methods to obtain $V^*$ as its unique solution.

### 2.1 Reinforcement learning with a generative model

In many settings the transition probabilities of the MRP are unknown, and the value function must be estimated based on data comprising sampled transitions, introducing a statistical element to the problem. A commonly used model for this data is a *generative model* (Kearns et al., 2002; Kakade, 2003). In this setting, for each state $x \in \mathcal{X}$, we observe $N$ i.i.d. samples $(X_i^x)_{i=1}^N$ from $P(\cdot | x)$, and this collection of $N|\mathcal{X}|$ samples may then be used by an algorithm to estimate the value function. Azar et al. (2013) showed that at least $N = \Omega(\varepsilon^{-2}(1-\gamma)^{-3} \log(|\mathcal{X}|/\delta))$ samples are required to obtain $\varepsilon$-accurate estimates of the value function with high probability (measured in $L^\infty$ norm), and also showed that this bound is attained (up to logarithmic factors) by a *certainty equivalence* algorithm, which treats the empirically observed transition frequencies as the true ones, and solves for the value function of the corresponding MRP.

### 2.2 Distributional reinforcement learning

Distributional RL aims to capture the full probability distribution of the random return at each state, not just its mean. Mathematically, the object of interest is the return-distribution function (RDF) $\eta^* : \mathcal{X} \to \mathscr{P}(\mathbb{R})$, defined by

$$\eta^*(x) = \mathcal{D}\left( \sum_{t=0}^{\infty} \gamma^t R_t \mid X_0 = x \right), \tag{3}$$

where $\mathcal{D}$ extracts the probability distribution of the input random variable. The distributional perspective on reinforcement learning has proved practically useful in a wide variety of applications, including healthcare (Böck et al., 2022), navigation (Bellemare et al., 2020), and algorithm discovery (Fawzi et al., 2022). The central equation behind dynamic programming approaches to approximating the return distribution function is the *distributional Bellman equation* (Sobel, 1982; Morimura et al., 2010a; Bellemare et al., 2017), given by $\eta^* = \mathcal{T}\eta^*$, where $\mathcal{T} : \mathscr{P}(\mathbb{R})^{\mathcal{X}} \to \mathscr{P}(\mathbb{R})^{\mathcal{X}}$ is the *distributional Bellman operator*, defined by

$$(\mathcal{T}\eta)(x) = \mathcal{D}\left( R + \gamma G(X') \mid X = x \right),$$

where independent from the random transition $(X, R, X')$, we have $G(x) \sim \eta(x)$ for each $x \in \mathcal{X}$.

## 2.3 Categorical dynamic programming

Given an initial RDF approximation $\eta \in \mathscr{P}([0, (1-\gamma)^{-1}])^{\mathcal{X}}$, it also holds that the update $\eta \leftarrow \mathcal{T}\eta$ converges to $\eta^*$ in an appropriate sense (e.g., in Wasserstein distance; see Bellemare et al., 2017), in analogy with dynamic programming algorithms for the value function, as described above. However, generally it is not possible to tractably implement repeated computation of the update $\eta \leftarrow \mathcal{T}\eta$ as a means of computing approximations to return distributions; probability distributions are infinite-dimensional objects, and as such computational costs quickly become prohibitive. Instead, the use of some kind of approximate, tractable representation of probability distributions is typically required.

**Representations.** In this paper, we focus on the categorical approach to distributional reinforcement learning (Bellemare et al., 2017), in which estimates of return distributions are represented as categorical distributions over a finite number of outcomes $z_1 < \cdots < z_m$. We will take $\{z_1, \ldots, z_m\}$ to be an equally spaced grid over the range of possible returns $[0, (1-\gamma)^{-1}]$, so that $z_i = \frac{i-1}{m-1}(1-\gamma)^{-1}$ for $i = 1, \ldots, m$. Approximations of the RDF are then represented in the form

$$\eta(x) = \sum_{i=1}^{m} p_i(x)\delta_{z_i}. \tag{4}$$

Here, $\delta_z$ is the Dirac distribution at the outcome $z$, and $p = ((p_i(x))_{i=1}^{m} : x \in \mathcal{X})$ are adjustable probability mass parameters; see Figure 1(a). The number of categories $m$ can be interpreted as controlling the *expressivity* of the representation, and should be carefully chosen in practice to trade off between increased accuracy (larger $m$), and computational tractability (smaller $m$).

**Dynamic programming.** The iteration $\eta \leftarrow \mathcal{T}\eta$ *cannot* be used to update the parameters $p$ in Equation (4) directly, since the distributions $(\mathcal{T}\eta)(x)$ are no longer supported on $\{z_1, \ldots, z_m\}$, and so cannot be expressed in the form given in Equation (4); see Figure 1(b). Bellemare et al. (2017) circumvent this issue by *projecting* the resulting distributions back onto the support set $\{z_1, \ldots, z_m\}$ via a map $\Pi_m : \mathscr{P}([0, (1-\gamma)^{-1}]) \to \mathscr{P}(\{z_1, \ldots, z_m\})$. Intuitively, $\Pi_m$ can be thought of as allocating each outcome $z \in [z_i, z_{i+1}]$ to its neighbouring gridpoints $z_i$ and $z_{i+1}$, in proportion to their proximity, so that the projection of the Dirac distribution $\delta_z$, is defined by

$$\Pi_m \delta_z = \frac{z_{i+1} - z}{z_{i+1} - z_i}\delta_{z_i} + \frac{z - z_i}{z_{i+1} - z_i}\delta_{z_{i+1}}.$$

In this paper, we work with the equivalent definition of the projection $\Pi_m$ given by Rowland et al. (2018, Proposition 6), in which the probability mass assigned to $z_i$ by $\Pi_m \nu$ is given by the expectation $\mathbb{E}_{Z \sim \nu}[h_i(Z)]$, where $h_i : [0, (1-\gamma)^{-1}] \to [0, 1]$ is the "hat function" at $z_i$, which linearly interpolates between a value of 1 at $z_i$, and 0 at neighbouring gridpoints $z_{i-1}, z_{i+1}$, and is 0 outside this range. Figure 1(c) illustrates $h_i$ and $h_m$; see Appendix B for a restatement of the full definition given by Rowland et al. (2018).

The projected update $\eta \leftarrow \Pi_m \mathcal{T}\eta$ is thus guaranteed to keep $\eta$ in the space of approximations of the form given in Equation (4), and can be viewed as a tractable alternative to the update $\eta \leftarrow \mathcal{T}\eta$ described above. Rowland et al. (2018) show that despite the introduction of the additional projection map $\Pi_m$, repeated computation of the update $\eta \leftarrow \Pi_m \mathcal{T}\eta$, referred to as *categorical dynamic programming* (CDP), is guaranteed to convergence to a *categorical fixed point*, and further, the categorical fixed point can be made arbitrarily close to the true RDF $\eta^*$ by increasing $m$, as measured by Cramér distance (Cramér, 1928; Székely, 2003; Székely and Rizzo, 2013).

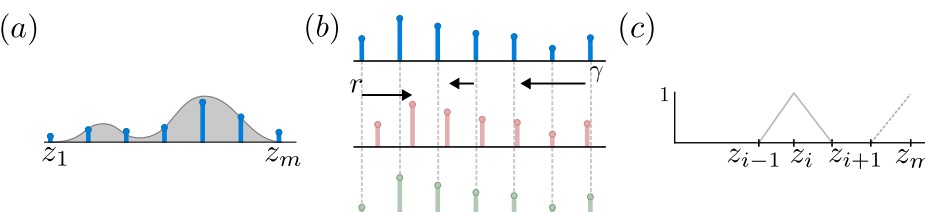

Figure 1: (a) The density of a distribution $\nu$ (grey), and its categorical projection $\Pi_m \nu \in \mathscr{P}(\{z_1, \ldots, z_m\})$ (blue). (b) A categorical distribution (blue); its update after being scaled by $\gamma$ and shifted by $r$ by the distributional Bellman operator $\mathcal{T}$, moving its support off the grid $\{z_1, \ldots, z_m\}$ (pink); the resulting realigned distribution supported on the grid $\{z_1, \ldots, z_m\}$ after projection via $\Pi_m$ (green). (c) Hat functions $h_i$ (solid) and $h_m$ (dashed).

**Definition 2.1.** The *Cramér distance* $\ell_2 : \mathscr{P}(\mathbb{R}) \times \mathscr{P}(\mathbb{R}) \to \mathbb{R}$ is defined by

$$\ell_2(\nu, \nu') = \left[ \int_{\mathbb{R}} (F_\nu(t) - F_{\nu'}(t))^2 \, \mathrm{d}t \right]^{1/2},$$

where $F_\nu, F_{\nu'}$ are the CDFs of $\nu, \nu'$, respectively. The *supremum-Cramér distance* $\bar{\ell}_2$ on $\mathscr{P}(\mathbb{R})^{\mathcal{X}}$ is defined by

$$\bar{\ell}_2(\eta, \eta') = \max_{x \in \mathcal{X}} \ell_2(\eta(x), \eta'(x)).$$

The central convergence results concerning CDP are summarised below.

**Proposition 2.2.** *(Rowland et al., 2018). The operator* $\Pi_m \mathcal{T} : \mathscr{P}([0, (1-\gamma)^{-1}])^{\mathcal{X}} \to \mathscr{P}([0, (1-\gamma)^{-1}])^{\mathcal{X}}$ *is a contraction mapping with respect to* $\bar{\ell}_2$, *with contraction factor* $\sqrt{\gamma}$, *and has a unique fixed point,* $\eta_C \in \mathscr{P}(\{z_1, \ldots, z_m\})^{\mathcal{X}}$. *As a result, for any* $\eta_0 \in \mathscr{P}([0, (1-\gamma)^{-1}])^{\mathcal{X}}$, *with* $\eta_{k+1} = \Pi_m \mathcal{T} \eta_k$, *we have* $\bar{\ell}_2(\eta_k, \eta_C) \leq (1-\gamma)^{-1} \gamma^k$. *Further, the distance between* $\eta_C$ *and the true RDF* $\eta^*$ *can be bounded as*

$$\bar{\ell}_2(\eta_C, \eta^*) \leq \frac{1}{(1-\gamma)\sqrt{m-1}}. \tag{5}$$

This establishes CDP as a principled approach to approximating return distributions, and also quantifies the accuracy achievable with CDP using $m$ categories, which will be central in informing our choice of $m$ to obtain a sample-efficient, accurate algorithm below.

# 3 Distributional reinforcement learning with a generative model

The central problem we study in this paper is how to do sample-efficient distributional RL with a generative model. That is, given the samples $((X_i^x)_{i=1}^N : x \in \mathcal{X})$ described in Section 2.1, how accurate of an approximation to the return-distribution function in Equation (3) can one compute?

This question was raised by Zhang et al. (2023), who proposed to perform distributional dynamic programming updates $\eta \leftarrow \hat{\mathcal{T}} \eta$ as described in Section 2.2, using the *empirical* distributional Bellman operator $\hat{\mathcal{T}}$ derived from the empirical transition probabilities $\hat{P}$, defined by $\hat{P}(y|x) = N^{-1} \sum_{i=1}^N \mathbb{1}\{X_i^x = y\}$, producing an estimate $\hat{\eta} \in \mathscr{P}([0, (1-\gamma)^{-1}])$ of the true RDF $\eta^*$ such that for any $\varepsilon > 0$ and $\delta \in (0, 1)$, we have $w_1(\hat{\eta}(x), \eta^*(x)) \leq \varepsilon$ with probability at least $1 - \delta$ for all $x \in \mathcal{X}$, whenever $N = \widetilde{\Omega}(\varepsilon^{-2}(1-\gamma)^{-4} \text{polylog}(1/\delta))$. Here, $w_1$ denotes the Wasserstein-1 distance between probability distributions, defined for any $\nu, \nu' \in \mathscr{P}(\mathbb{R})$ with CDFs $F_\nu, F_{\nu'}$ by $w_1(\nu, \nu') = \int_{\mathbb{R}} |F_\nu(t) - F_{\nu'}(t)| \, \mathrm{d}t$. We focus on the Wasserstein-1 distance as the main metric of interest in this paper as it is particularly compatible with existing methods for analysing categorical approaches to distributional RL, and it provides upper bounds for differences of many statistical functionals of interest, such as expectations of Lipschitz functions (Villani, 2009; Peyré and Cuturi, 2019); and conditional-value-at-risk (Rockafellar and Uryasev, 2000, 2002; Bhat and Prashanth, 2019, CVaR). Zhang et al. (2023) also prove a lower bound of $N = \widetilde{\Omega}(\varepsilon^{-2}(1-\gamma)^{-3})$ samples required to obtain such an accurate prediction with high probability, which follows from a reduction to the mean-return case (Azar et al., 2013).

There are two natural questions that the analysis of Zhang et al. (2023) leaves open. Firstly, can the gap between the lower bound and upper bound as a function of $(1-\gamma)^{-1}$ described above be closed? Zhang et al. (2023) conjecture that their analysis is loose, and that this gap can indeed be closed. Second, we also note that the distributional dynamic programming procedure $\eta \leftarrow \hat{\mathcal{T}} \eta$ proposed by Zhang et al. (2023), without incorporating any restrictions on the representations of distributions, runs into severe space and memory issues, and is not practical to run (and indeed Zhang et al. (2023) introduce approximations to the algorithm when running empirically for these reasons). A remaining question is then whether there are tractable algorithms that can achieve the lower bound on sample complexity described above. Our contributions below provide a new, tractable distributional RL algorithm that attains (up to logarithmic factors) the lower bound on sample complexity provided by Zhang et al. (2023), resolving these questions.

# 4 Direct categorical fixed-point computation

Our approach to obtaining a sample-efficient algorithm that is near-minimax-optimal in the sense described above begins with the categorical approach to distributional dynamic programming described in Section 2.3. We begin first by introducing a new algorithm for computing the categorical fixed point $\eta_C$ referred to in Proposition 2.2 *directly*, that avoids computing an approximate solution via dynamic programming iterations $\eta \leftarrow \Pi_m \mathcal{T} \eta$. We expect this algorithm to be of independent interest within the field of distributional reinforcement learning.

## 4.1 Direct categorical fixed-point computation

Our first contribution is to develop a new computational perspective on the projected categorical Bellman operator $\Pi_m \mathcal{T}$, which results in a new algorithm for computing the fixed point $\eta_C$ *exactly*, without requiring the iterative CDP algorithm described in Section 2.3.

**CDF operator and fixed-point equation.** We first formulate the application of the projected categorical Bellman operator $\Pi_m \mathcal{T}$ as a linear map, and give an explicit expression for the matrix representing this linear map when the input RDFs are represented with cumulative distribution functions (CDFs). We consider the effect of applying $\Pi_m \mathcal{T}$ to an RDF $\eta = \mathscr{P}(\{z_1, \ldots, z_m\})^{\mathcal{X}}$, with $\eta(x) = \sum_{i=1}^m p_i(x)\delta_{z_i}$. By Rowland et al. (2018, Proposition 6), the updated probabilities for $(\Pi_m \mathcal{T} \eta)(x) = \sum_{i=1}^m p_i'(x)\delta_{z_i}$ can be expressed as

$$p_i'(x) = \sum_{y \in \mathcal{X}} \sum_{j=1}^m P(y|x)h_{i,j}^x p_j(y) \,, \tag{6}$$

where $h_{i,j}^x = h_i(r(x) + \gamma z_j)$. We convert this into an expression for cumulative probabilities, rather than individual probability masses, to obtain a simpler analysis below. To do so, we introduce the encoding of $\eta \in \mathscr{P}(\{z_1, \ldots, z_m\})^{\mathcal{X}}$ into corresponding CDF values $F \in \mathbb{R}^{\mathcal{X} \times m}$, where $F_i(x) = \eta(x)([z_1, z_i])$ denotes the *cumulative* mass at state $x$ over the set $\{z_1, \ldots, z_i\}$.

**Proposition 4.1.** *If $\eta \in \mathscr{P}(\{z_1, \ldots, z_m\})^{\mathcal{X}}$ is an RDF with corresponding CDF values $F \in \mathbb{R}^{\mathcal{X} \times m}$, then the corresponding CDF values $F' \in \mathbb{R}^{\mathcal{X} \times m}$ for $\Pi_m \mathcal{T} \eta$ satisfy*

$$F_i'(x) = \sum_{y \in \mathcal{X}} \sum_{j=1}^m P(y|x)(H_{i,j}^x - H_{i,j+1}^x)F_j(y) \,, \tag{7}$$

*where*

$$H_{i,j}^x = \sum_{l \leq i} h_l(r(x) + \gamma z_j) \tag{8}$$

*for $j = 1, \ldots, m$, and by convention we take $H_{i,m+1}^x = 0$.*

Under this notation, we can rewrite Equation (7) simply as a matrix-vector multiplication in $\mathbb{R}^{\mathcal{X} \times [m]}$:

$$F' = T_P F \,,$$

where $T_P$ is the $(\mathcal{X} \times [m]) \times (\mathcal{X} \times [m])$ square matrix, with entries given by

$$T_P(x, i; y, j) = P(y|x)(H_{i,j}^x - H_{i,j+1}^x) \,, \tag{9}$$

and $F, F' \in \mathbb{R}^{\mathcal{X} \times m}$ above are interpreted in vectorised form. We drop dependence on $m$ from the notation $T_P$ for conciseness. Thus, CDP can be implemented via simple matrix multiplication on CDF values, and the CDF values $F^*$ for the categorical fixed point $\eta_C$ solve the equation

$$F = T_P F \,, \text{ or equivalently } (I - T_P)F = 0 \,. \tag{10}$$

**Obtaining a system with unique solution.** Equation (10) suggests that we can directly solve a linear system to obtain the exact categorical fixed point, rather than performing the iterative CDP algorithm to obtain an approximation. Note, however, that $F^*$ is not the *unique* solution of Equation (10); for example, $F = 0$ is also a solution. This arises because in Equation (10), the *distribution masses* at each state (that is, $F_m(x)$), are unconstrained. By contrast, Proposition 2.2 establishes that $\eta_C$ is the unique solution of $\eta = \Pi_m \mathcal{T} \eta$ in the space $\mathscr{P}([z_1, z_m])^{\mathcal{X}}$, where each element $\eta(x)$ is constrained

to be a probability distribution *a priori*. Thus, Equation (10) requires some modification to obtain a linear system with a unique solution. This is obtained by removing $F_m(x)$ as a variable from the system (for each $x \in \mathcal{X}$)), replacing it by the constant 1, and removing redundant rows from the resulting linear system, as the following proposition describes; the "axis-aligned" nature of these constraints is the benefit of working with CDF values.

**Proposition 4.2.** *The linear system in Equation* (10)*, with the additional linear constraints* $F_m(x) = 1$ *for all* $x \in \mathcal{X}$*, is equivalent to the following linear system in* $\widetilde{F} \in \mathbb{R}^{\mathcal{X} \times [m-1]}$*:*

$$(I - \widetilde{T}_P)\widetilde{F} = \widetilde{H} , \tag{11}$$

*where the* $(x, i; y, j)$ *coordinate of* $\widetilde{T}_P$ *(for* $1 \leq i, j \leq m - 1$*) is*

$$\widetilde{T}_P(x, i; y, j) = P(y|x)(H_{i,j}^x - H_{i,j+1}^x) ,$$

*and for each* $x \in \mathcal{X}$*,* $1 \leq i \leq m - 1$*, we have* $\widetilde{H}(x, i) = H_{i,m}^x$*.*

Having moved to the inhomogeneous system over $\mathbb{R}^{\mathcal{X} \times [m-1]}$ in Equation (11), we can deduce the following via the contraction theory in Proposition 2.2.

**Proposition 4.3.** *The linear system in Equation* (11) *has a unique solution, which is precisely the CDF values* $((F_i^*(x))_{i=1}^{m-1} : x \in \mathcal{X})$ *of the categorical fixed point.*

The *direct categorical fixed-point algorithm* (DCFP) consists of solving the linear system in Equation (11) to obtain the exact categorical fixed point; see Algorithm 1 for a summary, and Appendix G.5 for more details on implementations.

---

**Algorithm 1:** The direct categorical fixed-point algorithm (DCFP).

1  Calculate matrices $(H^x : x \in \mathcal{X})$ via Equation (8).
2  Calculate matrix $\widetilde{T}_P$ and vector $\widetilde{H}$ via Equation (9).
3  Call linear system solver on Equation (11).
4  Obtain resulting solution $\widetilde{F}^* \in \mathbb{R}^{\mathcal{X} \times [m-1]}$.
5  Return $F^*$, obtained by appending the values $F_m^*(x) = 1$ to the solution $\widetilde{F}^*$.

---

**Complexity and implementation details.** Representing $\widetilde{T}_P$ as a dense matrix requires $O(|\mathcal{X}|^2 m^2)$ space, and solving the corresponding linear system in Equation (11) with a standard linear solver requires $O(|\mathcal{X}|^3 m^3)$ time. However, in many problems there are cases where the DCFP algorithm can be implemented more efficiently. Crucially, $\widetilde{T}_P$ often has sparse structure, and so sparse linear solvers may afford an opportunity to make substantial improvements in computational efficiency. We explore this point further empirically in Section 6, and give a theoretical perspective in Appendix G.

### 4.2   DCFP with a generative model

We now return to the setting where the Markov reward process in which we are performing evaluation is unknown, and instead we have access to the random next-state samples $((X_i^x)_{i=1}^N : x \in \mathcal{X})$, as described in Section 3. Our model-based DCFP algorithm proceeds by first constructing the corresponding empirical transition probabilities $\hat{P}$, so that $\hat{P}(y|x) = N^{-1} \sum_{i=1}^N \mathbb{1}\{X_i^x = y\}$, and then calls the DCFP procedure outlined in Algorithm 1, treating $\hat{P}$ as the true transition probabilities of the MRP when constructing the matrix in Line 2, which we denote here by $T_{\hat{P}}$, to reflect the fact that it is built from $\hat{P}$. This produces the output CDF values $\hat{F}$, from which estimated return distributions can be decoded (with the convention $\hat{F}_0(x) = 0$) as

$$\hat{\eta}(x) = \sum_{i=1}^m (\hat{F}_i(x) - \hat{F}_{i-1}(x))\delta_{z_i} . \tag{12}$$

## 5   Sample complexity analysis

Our goal now is to analyse the sample complexity of model-based DCFP, as described in the previous section. We first introduce a notational shorthand that will be of extensive use in the statement and

proof of this result: when writing distances between distributions, such as $w_1(\eta^*(x), \hat{F}(x))$, we identify CDF vectors $\hat{F}(x) \in \mathbb{R}^m$ with the distributions they represent as in Equation (12). The core theoretical result of the paper is as follows.

**Theorem 5.1.** *Let $\varepsilon \in (0, (1-\gamma)^{-1/2})$ and $\delta \in (0, 1)$, and suppose the number of categories satisfies $m \geq 4(1-\gamma)^{-2}\varepsilon^{-2} + 1$. Then the output $\hat{F}$ of model-based DCFP with $N = \Omega(\varepsilon^{-2}(1-\gamma)^{-3}\mathrm{polylog}(|\mathcal{X}|/\delta))$ samples satisfies, with probability at least $1 - \delta$,*

$$\max_{x \in \mathcal{X}} w_1(\eta^*(x), \hat{F}(x)) \leq \varepsilon \,.$$

This result establishes that model-based DCFP attains the minimax lower-bound (up to logarithmic factors) for high-probability return distribution estimation in Wasserstein distance, resolving an open question raised by Zhang et al. (2023); in a certain sense, *estimation of return distributions is no more statistically difficult than that of mean returns with a generative model*. Note there is no direct dependence of $N$ on $m$, so there is no *statistical* penalty to using a large number of categories $m$.

**Extensions.** We note that this core result can be straightforwardly extended in several directions. In particular, similar bounds apply in the case of predicting returns for *learnt near-optimal policies* in MDPs (see Section F.2), for the *iterative categorical DP algorithm* in place of DCFP (when using sufficiently many DP updates; see Section F.1), and in the case of *stochastic rewards* (see Section F.3).

## 5.1 Structure of the proof of Theorem 5.1

The remainder of this section provides a sketch proof of Theorem 5.1; a complete proof is provided in the appendix. The proof is broadly motivated by the approaches of Azar et al. (2013), Agarwal et al. (2020), and Pananjady and Wainwright (2020), who analyse the mean-return case, and we highlight where key ideas and new mathematical objects are required in this distributional setting. In particular, we highlight the use of the *stochastic categorical CDF Bellman equation*, a new distributional Bellman equation that plays a key role in our analysis, which we expect to be of independent interest.

**Reduction to Cramér distance.** The first step of the analysis is to reduce Theorem 5.1 to a statement about approximation in Cramér distance, which is much better suited to the analysis of DCFP, owing to the results described in Section 2.3. This can be done by upper-bounding Wasserstein distance by Cramér distance using the following result, which is proven in the appendix via Jensen's inequality.

**Lemma 5.2.** *For any two distributions $\nu, \nu' \in \mathscr{P}([0, (1-\gamma)^{-1}])$, we have*

$$w_1(\nu, \nu') \leq (1-\gamma)^{-1/2}\ell_2(\nu, \nu') \,.$$

Theorem 5.1 is now reducible to the following, stated in terms of the Cramér distance.

**Theorem 5.3.** *Let $\varepsilon \in (0, 1)$ and $\delta \in (0, 1)$, and suppose the number of categories satisfies $m \geq 4(1-\gamma)^{-2}\varepsilon^{-2} + 1$. Then the output $\hat{F}$ of model-based DCFP with $N = \Omega(\varepsilon^{-2}(1-\gamma)^{-2}\mathrm{polylog}(|\mathcal{X}|/\delta))$ samples satisfies, with probability at least $1 - \delta$,*

$$\max_{x \in \mathcal{X}} \ell_2(\eta^*(x), \hat{F}(x)) \leq \varepsilon \,. \tag{13}$$

**Reduction to categorical fixed-point error.** Our first step in proving Theorem 5.3 is to use the triangle inequality to split the Cramér distance on the left-hand side of Equation (13) into a representation approximation error, and sample-based error:

$$\overline{\ell}_2(\eta^*, \hat{F}) \leq \overline{\ell}_2(\eta^*, F^*) + \overline{\ell}_2(F^*, \hat{F}) \leq \frac{1}{(1-\gamma)\sqrt{m-1}} + \overline{\ell}_2(F^*, \hat{F}) \,,$$

with the second inequality following from the fixed-point quality bound in Equation (5). With $m$ as specified in the theorem statement, the first term in the right-hand side above is bounded by $\varepsilon/2$. Thus, it suffices to focus on the second term on the right-hand side, which quantifies the sample-based error in estimating the categorical fixed point $F^*$.

**Concentration.** Through a combination of the use of a version of Bernstein's inequality in Hilbert space (Chatalic et al., 2022) and propagation of this inequality across time steps in the MRP, we next arrive at the following inequality with probability at least $1 - \delta$:

$$\ell_2(\hat{F}(x), F^*(x)) \leq \widetilde{O}\left(\frac{1}{\sqrt{N(1-\gamma)}}\sqrt{\|(I - \gamma\hat{P})^{-1}\sigma_{\hat{P}}\|_\infty} + \frac{1}{(1-\gamma)^{3/2}N^{3/4}}\right) \,. \tag{14}$$

Here, $\sigma_{\hat{P}} \in \mathbb{R}^{\mathcal{X}}$ is an instance of a new class of distributional object, the *local squared-Cramér variation*. The general definition is given below, for the case of a general transition matrix $Q \in \mathbb{R}^{\mathcal{X} \times \mathcal{X}}$, to avoid conflation with the specific transition matrices $P$ and $\hat{P}$ that Theorem 5.3 is concerned with.

**Definition 5.4.** For a given transition matrix $Q$, the *single-sample operator* $\hat{T}_Q : \mathbb{R}^{\mathcal{X} \times m} \to \mathbb{R}^{\mathcal{X} \times m}$ is the random operator given by: (i) constructing a *random transition matrix* $\hat{Q}$ by, for each $x \in \mathcal{X}$, sampling $X' \sim Q(\cdot|x)$, and setting $\hat{Q}(X'|x) = 1$; (ii) setting $\hat{T}_Q = T_{\hat{Q}}$.

**Definition 5.5.** For a given transition matrix $Q$ with corresponding CDP fixed point $F^Q \in \mathbb{R}^{\mathcal{X} \times m}$, the *local squared-Cramér variation at $Q$*, $\sigma_Q \in \mathbb{R}^{\mathcal{X}}$, is defined by

$$\sigma_Q(x) = \mathbb{E}[\ell_2^2((\hat{T}_Q F^Q)(x), F^Q(x))].$$

Intuitively, the local squared-Cramér variation $\sigma_Q$ encodes the variability of the fixed point $F^Q$ after a sample-based, rather than exact, dynamic programming update. From this point of view, it is a natural quantity to arise in Equation (14), and plays a similar role to the variance in the classical Bernstein inequality (Bernstein, 1946).

In Corollary 5.12 below, we will deduce that under the conditions of Theorem 5.3, we have

$$\|(I - \gamma \hat{P})^{-1} \sigma_{\hat{P}}\|_\infty \leq \frac{2}{1 - \gamma}. \tag{15}$$

Substituting this into Equation (14) gives

$$\ell_2(\hat{F}(x), F^*(x)) \leq \widetilde{O}\left(\frac{1}{(1 - \gamma)\sqrt{N}} + \frac{1}{(1 - \gamma)^{3/2} N^{3/4}}\right)$$

with probability at least $1 - \delta$, for all $x \in \mathcal{X}$. Now, taking $N = \widetilde{\Omega}((1 - \gamma)^{-2} \varepsilon^{-2})$ yields that this expression is $O(\varepsilon)$, which completes the sketch proof of Theorem 5.3. What remains to be described is how to arrive at the bound in Equation (15); the section below provides a high-level overview of the technical details involved in obtaining it.

## 5.2 The stochastic categorical CDF Bellman equation

The central idea is to relate the *local* squared-Cramér variation to a corresponding *global* notion of variation, in analogy with the variance Bellman equation (Sobel, 1982) used by Azar et al. (2013) in the mean-return case. To define this corresponding global notion, we begin by defining a new type of distributional Bellman equation, which can be intuitively thought of as encoding the result of repeatedly applying a sequence of independent single-sample operators. Again, we work with a general transition matrix $Q$.

**Definition 5.6.** For a general transition matrix $Q$, the *stochastic categorical CDF (SC-CDF) Bellman equation* is given by

$$\Phi(x) \stackrel{\mathcal{D}}{=} (\hat{T}_Q \Phi)(x), \tag{16}$$

where $\stackrel{\mathcal{D}}{=}$ denotes equality in distribution. Here, $\hat{T}_Q$ is a *single-sample operator* with respect to $Q$, as in Definition 5.4. Each $\Phi(x)$ is a random variable taking values in the space of valid CDF values $\mathscr{F} = \{F \in \mathbb{R}^m : 0 \leq F_1 \leq \cdots \leq F_{m-1} \leq F_m = 1\}$ for distributions in $\mathscr{P}(\{z_1, \ldots, z_m\})$, and is taken to be independent of the random operator $\hat{T}_Q$.

The intuition is that "unravelling" Equation (16) should lead to a solution of the form

$$\Phi(x) \stackrel{\mathcal{D}}{=} \lim_{k \to \infty} \hat{T}_Q^{(k)} \cdots \hat{T}_Q^{(1)} F,$$

where $(\hat{T}_Q^{(i)})_{i=1}^k$ are independent single-sample operators, so that $\Phi(x)$ encodes the fluctuations due to repeated CDP updates with randomly-sampled transitions. To make this intuition precise, we first verify that the SC-CDF Bellman equation has a unique solution.

**Proposition 5.7.** *The SC-CDF Bellman equation in Equation* (16) *has a unique solution, in the sense that there is a unique distribution for each $\Phi(x)$ such that Equation* (16) *holds for each $x \in \mathcal{X}$.*

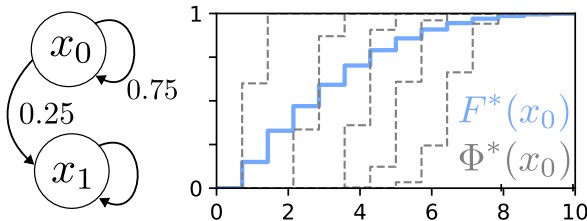

Figure 2: Left: Example MRP with $r(x_0) = 1, r(x_1) = 0, \gamma = 0.9$. Right: Categorical fixed point $F^*(x_0)$ with $m = 15$, and 5 independent samples from the random CDF $\Phi^*(x_0)$.

We write $\Phi^Q$ for the collection of random CDFs that satisfy the SC-CDF Bellman equation in Equation (16), whose existence is guaranteed by Proposition 5.7. We next relate $\Phi^Q$ to the standard categorical fixed point $F^Q$.

**Proposition 5.8.** *For all $x \in \mathcal{X}$, we have*

$$\mathbb{E}[\Phi^Q(x)] = F^Q(x).$$

Proposition 5.8 shows that $\Phi^Q$ can indeed thought of as encoding random variation around the usual categorical fixed point $F^Q$; see Figure 2. This motivates the following.

**Definition 5.9.** For a given transition matrix $Q$ with corresponding CDP fixed point $F^Q \in \mathbb{R}^{\mathcal{X} \times m}$, the *global squared-Cramér variation at $Q$, $\Sigma_Q \in \mathbb{R}^{\mathcal{X}}$*, is defined by

$$\Sigma_Q(x) = \mathbb{E}[\ell_2^2(\Phi^Q(x), F^Q(x))].$$

**Remark 5.10.** *Note that $\Phi^Q(x)$ is a* doubly *distributional object. It represents a probability distribution centred around the object $F^Q(x)$, which itself already provides a summary of the* distribution *of the return. This reveals a dual perspective on distributional RL itself. The distributional predictions made can serve several purposes: (i) modelling the aleatoric uncertainty in the return, as is the case in the work of Bellemare et al. (2017) and much subsequent algorithmic work, and/or (ii) serving to model specific types of epistemic uncertainty in the estimation of a non-random object from random data, as used in the analysis of Azar et al. (2013) and much subsequent work on the sample complexity of reinforcement learning. The object $\Phi^Q$ is motivated by* both *of these concerns simultaneously.*

The following Bellman-like inequality draws a relationship between local and global squared-Cramér variation, allowing us to make progress from Equation (15).

**Proposition 5.11.** *We have*

$$\Sigma_Q \geq \sigma_Q + \gamma Q \Sigma_Q - \left( \frac{2}{m\sqrt{1-\gamma}} + \frac{1}{m^2(1-\gamma)^2} \right) \mathbf{1},$$

*where $\mathbf{1} \in \mathbb{R}^{\mathcal{X}}$ is a vector of ones, and the inequality above is interpreted component-wise.*

Rearrangement and bounding of the quantities in the inequality of Proposition 5.11, in the specific case $Q = \hat{P}$, yields the required inequality in Equation (15) that completes the proof of Theorem 5.3.

**Corollary 5.12.** *We can bound the term $\|(I - \gamma\hat{P})^{-1}\sigma_{\hat{P}}\|_\infty$ appearing in Equation (14) under the assumptions of Theorem 5.3 as follows:*

$$\|(I - \gamma\hat{P})^{-1}\sigma_{\hat{P}}\|_\infty \leq \|\Sigma_{\hat{P}}\|_\infty + \frac{1}{1-\gamma} \leq \frac{2}{1-\gamma}.$$

## 6 Empirical evaluation

To complement our theoretical analysis, which focuses on worst-case sample complexity bounds, we report empirical findings for implementations of several model-based distributional RL algorithms in the generative model setting. We compare the new DCFP algorithm introduced in Section 4.1 with *quantile dynamic programming* (Dabney et al., 2018b; Bellemare et al., 2023, QDP) a distinct approach to distributional RL in which return distributions are approximated via dynamic programming with a finite number of quantiles.

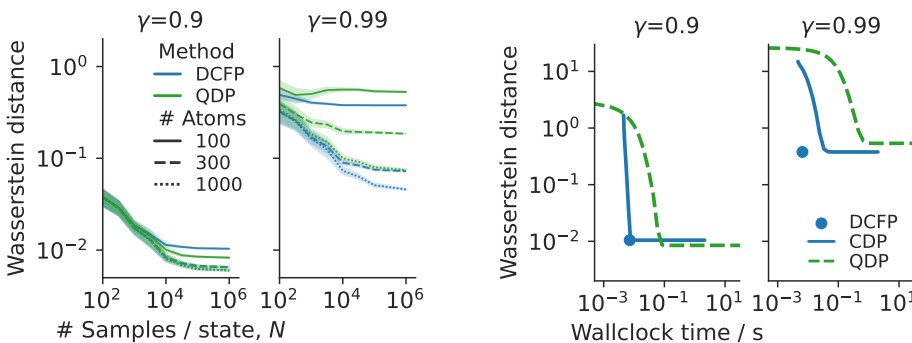

Figure 3: Approximation error/wallclock time for a variety of distributional RL methods, discount factors, numbers of atoms, and numbers of environment samples.

In Figure 3 (left), we report results of running DCFP and QDP on a 5-state environment with transition matrix randomly sampled from Dirichlet distributions, and entries of the immediate reward function $r \in \mathbb{R}^{\mathcal{X}}$ randomly sampled from $\mathrm{Unif}([0, 1])$, with varying numbers of atoms $m$, environment samples per state $N$, and discount $\gamma$. We report the maximum $w_1$-error against true return distributions (estimated via Monte Carlo sampling). All runs are repeated 30 times, and error bars are 95% bootstrapped confidence intervals. Sufficient DP iterations ensure approximate convergence to their fixed points. In Figure 3 (right), we plot estimation error against wallclock time for $m = 100, N = 10^6$, including results for CDP (which approximates the solution of DCFP via dynamic programming; Section 2.3); line plots indicate the estimation error/wallclock time trade-off as we increase the number of DP iterations. Both DCFP and CDP methods benefit from setting the atom support based on maximal/minimal values of $r$, as described in Appendix G.

For low discount factors and atom counts QDP generally outperforms DCFP in terms of asymptotic estimation error, due to QDP's ability to modify its atom support to regions of the interval $[0, (1 - \gamma)^{-1}]$ where mass is concentrated. However, we note that DCFP is generally faster than QDP. Further, DCFP generally outperforms QDP, in terms of both speed and estimation error, under larger discounts and/or larger atom count. We also note that particularly at high discounts, DCFP outperforms CDP in terms of wallclock time, due to DP methods requiring many iterations to converge in these cases (Rowland et al., 2018). Full results, including on several further environments, are given in Appendix G: key findings include that QDP works particularly well in near-deterministic environments, and DCFP works particularly well in settings where there are short high-probability paths from a state to itself.

## 7 Conclusion

We have introduced a new algorithm, DCFP, for directly computing the fixed point of CDP, a widely used distributional reinforcement learning algorithm. We then showed that this algorithm, with an appropriately chosen number of categories $m$, achieves the minimax lower bound (up to logarithmic factors) for sample complexity of return-distribution estimation in Wasserstein distance with a generative model. Thus, this paper closes an open question raised by Zhang et al. (2023) by exhibiting an algorithm that obtains this lower bound, and shows that estimation of return distributions via a generative model is essentially no harder statistically than the task of estimating a value function.

Our analysis also casts new light on categorical approaches to distributional reinforcement learning in general. The newly introduced stochastic categorical CDF Bellman equation serves to encode information about statistical fluctuations of categorical approaches to distributional RL, and we expect it to be of further use in theoretical work in distributional RL generally. Our experimental results also highlight salient differences in performance for distributional RL algorithms making use of distinct representations, depending on levels of environment stochasticity and discount factor in particular. We believe further investigation of these phenomena is an interesting direction for future work.

## Acknowledgments and Disclosure of Funding

We thank Dave Abel for detailed comments on a draft version of the paper. We also thank Mohammad Gheshlaghi Azar for useful conversations, and Arthur Gretton for advice on Bernstein-like inequalities in Hilbert space.

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

# APPENDICES:
# Near-Minimax-Optimal Distributional Reinforcement Learning
# with a Generative Model

For convenience, we provide a summary of the contents of the appendices below.

- Appendix A provides a detailed discussion of related work.
- Appendix B provides additional convenient notation for the categorical CDF operator $T_P$, in particular expressing it as the combination of a scaling/shifting/projecting operation, and a mixing operation over the state to be bootstrapped from. Several additional self-contained contraction results are given that are used within the main proofs of the paper.
- Appendix C provides proofs for the results in Section 4 concerning the formation of the linear system that is solved by the DCFP algorithm, and verification that the derived system has the unique desired solution.
- Appendix D provides proofs relating to the stochastic categorical CDF Bellman equation, in particular establishing the desired unique solution to this new distributional Bellman equation, and proving a key bound on the local squared-Cramér variation that is used within the proof of the main theorem of the paper.
- Appendix E provides the proof of the main result of the paper, Theorem 5.1, giving full details for the steps traced through in the sketch proof in the main paper.
- Appendix F provides discussion of several straightforward extensions of the main result, Theorem 5.1.
- Appendix G provides full details relating to the experiments in the main paper, as well as additional empirical comparisons between methods for distributional RL with a generative model.

## A  Related work

**Other families of model-based distributional RL algorithms.** There are many approaches to distributional RL; important choices studied previously include moments (Sobel, 1982), exponential families (Morimura et al., 2010b), categorical distributions (Bellemare et al., 2017), collections of particles (Morimura et al., 2010a; Nguyen-Tang et al., 2021), quantiles (Dabney et al., 2018b), and generative models (Doan et al., 2018; Dabney et al., 2018a; Freirich et al., 2019; Yang et al., 2019; Yue et al., 2020). Our choice of categorical representations in this work is motivated by several considerations: (i) the existence of principled dynamic programming methods for these representations, with corresponding convergence theory (Rowland et al., 2018); (ii) the flexibility of these representations to trade-off computational complexity with accuracy (by varying $m$) (Rowland et al., 2018); (iii) the mathematical structure of the dynamic programming operator (linear), as described in this work; and (iv) the availability of an efficient algorithm to exactly compute the DP fixed point (the DCFP algorithm proposed in Section 4.1).

An interesting and important direction, given our empirical findings in Section 6, is whether analyses can also be carried out for other approaches to distributional dynamic programming, such as quantile dynamic programming (QDP; Dabney et al., 2018b; Rowland et al., 2024; Bellemare et al., 2023), and fitted likelihood estimation (FLE; Wu et al., 2023). We expect challenges in extending the analysis to result due to the fact that, for example, the QDP operator is non-linear, and FLE operator applications typically do not have closed forms. Nevertheless, it would be particularly interesting to understand whether it is possible to obtain instance-dependent bounds for QDP, particularly given its strong empirical performance. Similarly, as described in Section 5, an interesting question for future work is whether it is possible to improve on the computational complexity of model-based DCFP for high-probability return-direction estimation.

**Other statistical questions in distributional RL.** Böck and Heirzinger (2022) propose a *model-free* algorithm for distributional RL, speedy categorical TD-learning, motivated by categorical TD learning (Rowland et al., 2018) and speedy Q-learning (Azar et al., 2011), and prove a sample complexity bound of $\widetilde{O}(\varepsilon^{-2}(1-\gamma)^{-3})$ for high probability $\varepsilon$-accurate estimation in Cramér distance (which implies a non-minimax sample complexity of $\widetilde{O}(\varepsilon^{-2}(1-\gamma)^{-4})$ in Wasserstein-1 distance, per Lemma 5.2). Wu et al. (2023) study the offline evaluation problem, in which state-action pairs are sampled from the stationary distribution of the policy, via fitted likelihood estimation (FLE) and focus on generalisation bounds, allowing for policy evaluation in environments with uncountable state

spaces. Wang et al. (2023c) also study policy evaluation in this context, focusing on the LQR model. Aside from studying minimax optimality, Zhang et al. (2023) also make several other contributions on the topic of statistical efficiency of estimation for distributional RL, including analysis of asymptotic fluctuations and limit theorems, and analysis of approximations in more general metrics, including Wasserstein-$p$ , Kolmogorov-Smirnov, and total variation metrics. Their sample complexity results rely on careful analysis of the behaviour of the *unprojected* distributional Bellman operator $\mathcal{T}$ on certain subspaces of probability/signed measures.

**Sample complexity of mean-return estimation.** The sample complexity of estimating mean returns, and the related task of obtaining a near-optimal policy, has been considered by Azar et al. (2013); Sidford et al. (2018); Pananjady and Wainwright (2020); Agarwal et al. (2020); Li et al. (2020). Interestingly, Agarwal et al. (2020), while treating the mean-return case, make use of return-binning as a proof technique. Li et al. (2020) also obtain bounds for mean-return sample complexity that apply with $\varepsilon > (1-\gamma)^{-1/2}$ for modifications of certainty-equivalent model-based algorithms; an interesting direction for future work would be to check whether the restrictions on $\varepsilon$ in Theorem 5.1 can be lifted by incorporating ideas from this mean-return analysis to the distributional setting. Additionally, Chandak et al. (2021) consider the task of estimating the variance of returns from off-policy data, and Wang et al. (2023b) study regret minimisation properties (with respect to the expected return criterion) of distributional RL algorithms in the online setting.

**Risk-sensitive control.** The theory developed in this paper has focused on estimation of return distributions for individual policies. A natural direction for future work is to analyse identification of near-optimal policies for risk-sensitive decision criteria. Bastani et al. (2022); Wang et al. (2023a) study efficient algorithms for CVaR optimisation, while Fei et al. (2021a,b); Liang and Luo (2022) study entropic risk maximisation, and Du et al. (2022); Lam et al. (2022) study iterated CVaR optimisation, all in the *online* setting.

**Further analysis.** In this paper, we have resolved a conjecture of Zhang et al. (2023), obtaining a near-minimax-optimal algorithm for estimation of return distributions in Wasserstein-1 distance. Zhang et al. (2023) make contributions to several other important statistical questions regarding distributional RL, including approximation in stronger metrics such as Kolmogorov-Smirnov and total variation metrics, as well as studying asymptotic fluctuations of estimators; it will be interesting to see whether the analysis presented here can be extended to these other settings as well.

# B   Additional CDF operator notation and contractivity results

In this section, we introduce finer-grained notation for the CDF operator $T_P$ that allows us to straightforwardly refer to the operations that correspond to shifting/scaling/projecting, and to mixing over transition states, separately. We also establish several additional contraction lemmas that will be useful in the sections that follow.

## B.1   Categorical hat function definition

For convenience, we provide the full mathematical definition of the hat functions $h_i : [0, (1-\gamma)^{-1}] \to [0, 1]$ used in defining the categorical projection described in the main paper, as given by Rowland et al. (2018). For $i = 2, \ldots, m-1$, we have

$$h_i(z) = \begin{cases} \frac{z - z_{i-1}}{z_i - z_{i-1}} & \text{for } z \in [z_{i-1}, z_i] \\ \frac{z_{i+1} - z}{z_{i+1} - z_i} & \text{for } z \in [z_i, z_{i+1}] \\ 0 & \text{otherwise.} \end{cases}$$

For the edge case $h_1$, we have

$$h_1(z) = \begin{cases} \frac{z_2 - z}{z_2 - z_1} & \text{for } z \in [z_1, z_2] \\ 0 & \text{otherwise,} \end{cases}$$

and similarly for the edge case $h_m$, we have

$$h_m(z) = \begin{cases} \frac{z - z_{m-1}}{z_m - z_{m-1}} & \text{for } z \in [z_{m-1}, z_m] \\ 0 & \text{otherwise.} \end{cases}$$

## B.2 Finer-grained operator expression

Recall that the CDF operator $T_P : \mathbb{R}^{\mathcal{X} \times m} \to \mathbb{R}^{\mathcal{X} \times m}$ is represented by a matrix with elements given by

$$T_P(x, i; j, y) = P(y|x)(H_{i,j}^x - H_{i,j+1}^x). \tag{17}$$

We can therefore conceptualise the application $T_P F$ as standard matrix-vector multiplication in the vector space $\mathbb{R}^{\mathcal{X} \times m}$. However, the expression for matrix elements in Equation (17) has additional structure that means we can express the application of $T_P$ in a different manner, which will be convenient in several proofs below, particularly as it separates out the influence of rewards (which remain fixed) and transition dynamics (which are estimated via samples).

In particular, we will regard $F \in \mathbb{R}^{\mathcal{X} \times m}$ itself as a matrix, with rows indexed by states in $\mathcal{X}$, and columns indexed by indices $i = 1, \ldots, m$. For each state $x \in \mathcal{X}$, we then introduce the matrix $B_x \in \mathbb{R}^{m \times m}$, with $(i, j)$ element given by

$$H_{i,j}^x - H_{i,j+1}^x .$$

We then have that $(T_P F)(x) \in \mathbb{R}^m$ can alternatively be expressed in matrix notation as

$$P_x F B_x^\top ,$$

where $P_x$ is the row vector given by the row of $P$ corresponding to state $x \in \mathcal{X}$.

## B.3 Additional results

Below, we provide several additional results regarding contractivity properties of $T_P$. To do so, it is useful to introduce the norm $\| \cdot \|_{\ell_2}$ on $\mathbb{R}^m$, which we define by

$$\|F\|_{\ell_2} = \left[ \frac{1}{m(1-\gamma)} \sum_{i=1}^m F_i(x)^2 \right]^{1/2} .$$

The motivation for this definition is that if we have two distributions $\nu, \nu' \in \mathscr{P}(\{z_1, \ldots, z_m\})$ with corresponding CDF vectors $F, F' \in \mathbb{R}^m$, then $\ell_2(\nu, \nu') = \|F - F'\|_{\ell_2}$, as $\nu, \nu'$ are supported on $[0, (1 - \gamma)^{-1}]$. Thus, under the abuse of notation $\ell_2(F, F')$ introduced in the main paper, we have $\ell_2(F, F') = \|F - F'\|_{\ell_2}$. We also introduce a supremum version of this norm on the space $\mathbb{R}^{\mathcal{X} \times m}$, which we denote by $\| \cdot \|_{\ell_2, \infty}$, and define by

$$\|F\|_{\ell_2, \infty} = \max_{x \in \mathcal{X}} \|F(x)\|_{\ell_2} ,$$

for all $F \in \mathbb{R}^{\mathcal{X} \times m}$. This norm is defined so that if we have RDFs $\eta, \eta' \in \mathscr{P}(\{z_1, \ldots, z_m\})^{\mathcal{X}}$, and $F, F' \in \mathbb{R}^{\mathcal{X} \times m}$ are the corresponding CDF values, then

$$\bar{\ell}_2(\eta, \eta') = \|F - F'\|_{\ell_2, \infty} .$$

With this norm defined, we can now state and prove our first result, which essentially translates the contraction result in Proposition 2.2, expressed purely in terms of probability distributions and the Cramér distance $\ell_2$, into a slightly more general result expressed over $\mathbb{R}^{\mathcal{X} \times m}$ and the norm $\| \cdot \|_{\ell_2}$.

**Proposition B.1.** *The operator* $T_P : \mathbb{R}^{\mathcal{X} \times m} \to \mathbb{R}^{\mathcal{X} \times m}$ *is a contraction when restricted to the subspace* $\{F \in \mathbb{R}^{\mathcal{X} \times m} : F_m(x) = 0 \text{ for all } x \in \mathcal{X}\}$ *with respect to the norm* $\| \cdot \|_{\ell_2, \infty}$, *with contraction factor* $\sqrt{\gamma}$.

*Proof.* By Proposition 2.2, for any two RDF approximations $\eta, \eta' \in \mathscr{P}(\{z_1, \ldots, z_m\})^{\mathcal{X}}$ with corresponding CDF values $F, F' \in \mathbb{R}^{\mathcal{X}}$, we have

$$\bar{\ell}_2(\Pi_m \mathcal{T} \eta, \Pi_m \mathcal{T} \eta') \leq \sqrt{\gamma} \bar{\ell}_2(\eta, \eta') ,$$

and hence we also have

$$\|T_P F - T_P F'\|_{\ell_2, \infty} \leq \sqrt{\gamma} \|F - F'\|_{\ell_2, \infty} .$$

Hence, $T_P$ is a contraction map on the set

$$\{F - F' : 0 \leq F_1(x) \leq \cdots \leq F_m(x) = 1 , \, 0 \leq F_1'(x) \leq \cdots \leq F_m'(x) = 1 \text{ for all } x \in \mathcal{X}\} .$$

This set contains a basis for the subspace $\{F \in \mathbb{R}^{\mathcal{X} \times m} : F_m(x) = 0 \text{ for all } x \in \mathcal{X}\}$. Namely, the one-hot vector at coordinate $(x, j)$ (for $j < m$) can be exhibited as lying in this subspace since it can be expressed as the difference between the vectors $F, F' \in \mathbb{R}^{\mathcal{X} \times m}$ defined by $F_j(y) = F'_j(y) = 1$ for all $y \neq x$, and all $j = 1, \ldots, m$, and $F_i(x) = 1$ for $i \geq j$ (and 0 otherwise), and $F'_i(x) = 1$ for $i \geq j + 1$ (and 0 otherwise). Since $T_P$ is linear, it therefore holds that

$$\|T_P F\|_{\ell_2,\infty} \leq \sqrt{\gamma} \|F\|_{\ell_2,\infty}, \tag{18}$$

for any $F$ in the subspace $\{F \in \mathbb{R}^{\mathcal{X} \times m} : F_m(x) = 0 \text{ for all } x \in \mathcal{X}\}$, as required. □

**Proposition B.2.** *For each $x \in \mathcal{X}$, the matrix $B_x$ is a contraction mapping on the space $\{F \in \mathbb{R}^m : F_m = 0\}$ with respect to $\|\cdot\|_{\ell_2}$, with contraction factor $\sqrt{\gamma}$.*

*Proof.* Consider a related one-state MRP, for which the reward at the single state is $r(x)$, and the state transitions to itself with probability 1. The categorical Bellman operator associated with this MRP and the support $\{z_1, \ldots, z_m\}$ is precisely $B_x$, and the statement of the result therefore follows as a special case of Proposition B.1. □

The next result serves as a counterpoint to Proposition B.2; it shows that if $m$ is sufficiently large, the map $B_x$ does not contract by too much in $\|\cdot\|_{\ell_2}$.

**Proposition B.3.** *For any $F, F' \in \mathscr{F}$, we have*

$$\|B_x F - B_x F'\|_{\ell_2}^2 \geq \gamma \|F - F'\|_{\ell_2}^2 - \frac{2}{m(1-\gamma)^{1/2}} - \frac{1}{m^2(1-\gamma)^2}.$$

This is proven via the following lemma.

**Lemma B.4.** *For any $\nu \in \mathscr{P}([0, (1-\gamma)^{-1}])$ and the projection $\Pi_m : \mathscr{P}([0, (1-\gamma)^{-1}]) \to \mathscr{P}(\{z_1, \ldots, z_m\})$, we have*

$$\ell_2(\nu, \Pi_m \nu) \leq \frac{1}{2\sqrt{m(1-\gamma)}},$$

*Further, for any $\nu, \nu' \in \mathscr{P}([0, (1-\gamma)^{-1}])$, we have*

$$\ell_2(\Pi_m \nu, \Pi_m \nu') \geq \ell_2(\nu, \nu') - \frac{1}{m(1-\gamma)},$$

*and*

$$\ell_2^2(\Pi_m \nu, \Pi_m \nu') \geq \ell_2^2(\nu, \nu') - \frac{2}{m(1-\gamma)^{1/2}} - \frac{1}{m^2(1-\gamma)^2}.$$

*Proof.* By Rowland et al. (2018, Proposition 6), we have that the CDF values $F_{\Pi_m \nu}(z_i)$ for $i = 1, \ldots, m-1$ are equal to the average of the CDF values of $F_\nu$ on the interval $[z_i, z_{i+1}]$. Therefore, in computing the squared Cramér distance $\ell_2^2(\nu, \Pi_m \nu)$, the worst-case contribution to the integral

$$\ell_2^2(\nu, \Pi_m \nu) = \int_0^{(1-\gamma)^{-1}} (F_\nu(t) - F_{\Pi_m \nu}(t))^2 \, \mathrm{d}t$$

from the interval $[z_i, z_{i+1}]$, holding $F_\nu(z_i)$ and $F_\nu(z_{i+1})$ constant, is

$$(z_{i+1} - z_i) \left( \frac{F(z_{i+1}) - F(z_i)}{2} \right)^2 = \frac{1}{4m(1-\gamma)} (F(z_{i+1}) - F(z_i))^2.$$

Thus, the worst-case value for the entire integral is

$$\frac{1}{4m(1-\gamma)} \sum_{i=1}^{m-1} (F(z_i) - F(z_{i+1}))^2.$$

The worst-case value for the sum is 1, by interpreting this as a sum of squared probabilities, and we therefore deduce that

$$\ell_2^2(\nu, \Pi_m \nu) \leq \frac{1}{4m(1-\gamma)},$$

leading to

$$\ell_2(\nu, \Pi_m \nu) \leq \frac{1}{2\sqrt{m(1-\gamma)}} \,,$$

as required for the first stated inequality. For the second and third inequalities, we apply the triangle inequality twice to obtain

$$\ell_2(\nu, \nu') \leq \ell_2(\nu, \Pi_m \nu) + \ell_2(\Pi_m \nu, \Pi_m \nu') + \ell_2(\Pi_m \nu', \nu') \leq \ell_2(\Pi_m \nu, \Pi_m \nu') + \frac{1}{m(1-\gamma)} \,;$$

rearrangement then gives the second statement. Squaring both sides of the inequality above gives

$$\ell_2^2(\nu, \nu') \leq \ell_2^2(\Pi_m \nu, \Pi_m \nu') + \frac{2}{m(1-\gamma)} \ell_2(\Pi_m \nu, \Pi_m \nu') + \frac{1}{m^2(1-\gamma)^2} \,.$$

Bounding the instance of $\ell_2(\Pi_m \nu, \Pi_m \nu')$ in the cross-term on the right-hand side by $(1-\gamma)^{1/2}$ and rearranging then yields the result. $\qquad\square$

*Proof of Proposition B.3.* Let $\nu, \nu' \in \mathscr{P}(\{z_1, \ldots, z_m\})$ be the distributions with CDF values $F, F'$, respectively, and let $G, G'$ be random variables with CDFs $F, F'$ respectively, and $(x, X')$ an independent random transition in the MRP beginning at state $x$. Recall from the notation introduced earlier in this section that $B_x F$ and $B_x F'$ are the CDF values of the distributions of $r(x) + \gamma G$ and $r(x) + \gamma G'$ after projection onto the support grid $\{z_1, \ldots, z_m\}$ by the projection map $\Pi_m$. Following common notation in distributional RL (Bellemare et al., 2023), we denote the distributions of $r(x) + \gamma G$ and $r(x) + \gamma G'$ by $(b_{r(x),\gamma})_{\#}\nu$ and $(b_{r(x),\gamma})_{\#}\nu'$, respectively. Here, $b_{r(x),\gamma} : \mathbb{R} \to \mathbb{R}$ is the bootstrap function $b_{r(x),\gamma}(z) = r(x) + \gamma z$, and $(b_{r(x),\gamma})_{\#}\nu$ is the push-forward distribution of $\nu$ through $b_{r(x),\gamma}$ (intuitively, the distribution obtained by transforming the support of $\nu$ according to $b_{r(x),\gamma}$). With this notation introduced, we therefore have

$$
\begin{aligned}
\|B_x F - B_x F'\|_{\ell_2}^2 &= \ell_2^2(\Pi_m (b_{r(x),\gamma})_{\#}\nu, \Pi_m (b_{r(x),\gamma})_{\#}\nu') \\
&\overset{(a)}{\geq} \ell_2^2((b_{r(x),\gamma})_{\#}\nu, (b_{r(x),\gamma})_{\#}\nu') - \frac{2}{m(1-\gamma)^{1/2}} - \frac{1}{m^2(1-\gamma)^2} \\
&\overset{(b)}{=} \gamma \ell_2^2(\nu, \nu') - \frac{2}{m(1-\gamma)^{1/2}} - \frac{1}{m^2(1-\gamma)^2} \\
&= \gamma \|F - F'\|_{\ell_2}^2 - \frac{2}{m(1-\gamma)^{1/2}} - \frac{1}{m^2(1-\gamma)^2} \,,
\end{aligned}
$$

as required, where (a) follows from Lemma B.4, and (b) follows from homogeneity of Cramér distance (see Rowland et al. (2018, Proof of Proposition 2)). $\qquad\square$

# C   Proofs of results in Section 4

**Proposition 4.1.** *If $\eta \in \mathscr{P}(\{z_1, \ldots, z_m\})^{\mathcal{X}}$ is an RDF with corresponding CDF values $F \in \mathbb{R}^{\mathcal{X} \times m}$, then the corresponding CDF values $F' \in \mathbb{R}^{\mathcal{X} \times m}$ for $\Pi_m \mathcal{T} \eta$ satisfy*

$$F'_i(x) = \sum_{y \in \mathcal{X}} \sum_{j=1}^{m} P(y|x)(H^x_{i,j} - H^x_{i,j+1}) F_j(y) \,, \tag{7}$$

*where*

$$H^x_{i,j} = \sum_{l \leq i} h_l(r(x) + \gamma z_j) \tag{8}$$

*for $j = 1, \ldots, m$, and by convention we take $H^x_{i,m+1} = 0$.*

*Proof.* Beginning by restating Equation (6), we have that if $\eta \in \mathscr{P}(\{z_1, \ldots, z_m\})^{\mathcal{X}}$ is an RDF with corresponding probability mass values $p \in \mathbb{R}^{\mathcal{X} \times m}$, then the updated RDF $\eta' = \Pi_m \mathcal{T} \eta$ has corresponding probability mass values $p' \in \mathbb{R}^{\mathcal{X} \times m}$ given by

$$p'_l(x) = \sum_{y \in \mathcal{X}} \sum_{j=1}^{m} P(y|x) h^x_{l,j} p_j(y) \,.$$

First, for $i \in \{1, \ldots, m\}$, we sum $l$ from 1 to $i$ to yield

$$F_i'(x) = \sum_{l \leq i} p_l'(x) = \sum_{y \in \mathcal{X}} \sum_{j=1}^{m} P(y|x) \sum_{l \leq i} h_{l,j}^x p_j(y) = \sum_{y \in \mathcal{X}} \sum_{j=1}^{m} P(y|x) H_{i,j}^x p_j(y)$$

$$= \sum_{y \in \mathcal{X}} \sum_{j=1}^{m} P(y|x) H_{i,j}^x (F_j(y) - F_{j-1}(y)),$$

where by convention we take $F_0(y) \equiv 0$. By reorganising the terms on the right-hand side, the claim now follows. $\qquad\square$

**Proposition 4.2.** *The linear system in Equation* (10), *with the additional linear constraints* $F_m(x) = 1$ *for all* $x \in \mathcal{X}$, *is equivalent to the following linear system in* $\widetilde{F} \in \mathbb{R}^{\mathcal{X} \times [m-1]}$:

$$(I - \widetilde{T}_P)\widetilde{F} = \widetilde{H}, \tag{11}$$

*where the* $(x, i; y, j)$ *coordinate of* $\widetilde{T}_P$ *(for* $1 \leq i, j \leq m - 1$*) is*

$$\widetilde{T}_P(x, i; y, j) = P(y|x)(H_{i,j}^x - H_{i,j+1}^x),$$

*and for each* $x \in \mathcal{X}$, $1 \leq i \leq m - 1$, *we have* $\widetilde{H}(x, i) = H_{i,m}^x$.

*Proof.* First, we consider a row of $F = T_P F$ that corresponds to the index $(x, i)$, with $i \neq m$. Expanding under the definition of $T_P$, we have

$$F_i(x) = \sum_{y \in \mathcal{X}} \sum_{j=1}^{m-1} P(y|x)(H_{i,j}^x - H_{i,j+1}^x)F_j(y) + \sum_{y \in \mathcal{X}} P(y|x)H_{i,m}^x F_m(y).$$

Since we assume the additional constraints $F_m(y) \equiv 1$ for all $y \in \mathcal{X}$, the final term on the right-hand side can be simplified to yield

$$F_i(x) = \sum_{y \in \mathcal{X}} \sum_{j=1}^{m-1} P(y|x)(H_{i,j}^x - H_{i,j+1}^x)F_j(y) + H_{i,m}^x.$$

This is precisely the row of $\widetilde{F} = \widetilde{T}_P \widetilde{F} + \widetilde{H}$ corresponding to index $(x, i)$.

Now, we consider a row of $F = T_P F$ that corresponds to the index $(x, m)$. Again making the substitution $F_m(y) \equiv 1$ for all $y \in \mathcal{X}$, we have

$$1 \equiv F_m(x) = \sum_{y} \sum_{j=1}^{m-1} P(y|x) \overbrace{(H_{m,j}^x - H_{m,j+1}^x)}^{=0} F_j(y) + \sum_{y} P(y|x)H_{m,m}^x F_m(y)$$

$$= \sum_{y} P(y|x)H_{m,m}^x F_m(y) \equiv 1,$$

which shows that the equation is redundant, and hence can be removed from the system. The claim $H_{m,j}^x - H_{m,j+1}^x = 0$ follows since in fact $H_{m,j}^x = \sum_{i=1}^{m} h_i(r(x) + \gamma z_j)$, and the sum over the hat functions for any input argument is 1. In the final equality, we have used the fact that $H_{m,m}^x = 1$ similarly. Thus, we have deduced the claim of the proposition. $\qquad\square$

**Proposition 4.3.** *The linear system in Equation* (11) *has a unique solution, which is precisely the CDF values* $((F_i^*(x))_{i=1}^{m-1} : x \in \mathcal{X})$ *of the categorical fixed point.*

*Proof.* By Proposition 4.2, we have that the CDF values $\widetilde{F}^*$ of the categorical fixed-point solve Equation (11). Now, let us suppose that $\widetilde{F}_1, \widetilde{F}_2$ are distinct solutions to Equation (11), aiming to obtain a contradiction. Proposition 4.2 also establishes that Equation (11) is equivalent to Equation (10) with the additional conditions that $F_m(x) = 1$ for all $x \in \mathcal{X}$, so we will denote the corresponding two solutions to Equation (10) built from $\widetilde{F}_1, \widetilde{F}_2$ (by setting the unspecified $(x, m)$ coordinates to 1) by $F_1, F_2$, respectively.

The idea is now to use the contractivity of the projected operator $\Pi_m \mathcal{T}$ in $\ell_2$, as established in Proposition 2.2, to obtain a contradiction. Notationally, it is useful to phrase things in terms of contractivity of the CDF operator $T_P$ itself. We use the norms $\|\cdot\|_{\ell_2}$ and $\|\cdot\|_{\ell_2,\infty}$, defined on $\mathbb{R}^m$ and $\mathbb{R}^{\mathcal{X} \times m}$ in Appendix B, which we recall here for convenience:

$$\|F\|_{\ell_2} = \left[\frac{1}{m(1-\gamma)}\sum_{i=1}^{m}F_i^2\right]^{1/2} \quad \text{for } F \in \mathbb{R}^m, \quad \text{and} \quad \|F\|_{\ell_2,\infty} = \max_{x \in \mathcal{X}} \|F(x)\|_{\ell_2} \text{ for } F \in \mathbb{R}^{\mathcal{X} \times m}$$

We therefore have

$$\|T_P(F_1 - F_2)\|_{\ell_2,\infty} = \|T_P F_1 - T_P F_2\|_{\ell_2,\infty} = \|F_1 - F_2\|_{\ell_2,\infty}.$$

However, Proposition B.1 establishes that $T_P$ is a contraction on $\{F \in \mathbb{R}^{\mathcal{X} \times m} : F_m(x) = 0 \text{ for all } x \in \mathcal{X}\}$ with respect to $\|\cdot\|_{\ell_2,\infty}$, which contradicts the statement above, as required. $\square$

# D   Proofs relating to the stochastic CDF Bellman equation

Before giving the proofs, we present a version of the SC-CDF Bellman equation in a purely distributional form, which will streamline the arguments. This mirrors the development of the form of the distributional Bellman equation given purely in terms of distributions (Rowland et al., 2018), rather than in random variable form as in Bellemare et al. (2017). We define the stochastic categorical CDF Bellman operator $\mathcal{T}_{\text{SCC}} : \mathscr{P}(\mathscr{F})^{\mathcal{X}} \to \mathscr{P}(\mathscr{F})^{\mathcal{X}}$ for each $\psi \in \mathscr{P}(\mathscr{F})^{\mathcal{X}}$ by

$$(\mathcal{T}_{\text{SCC}} \, \psi)(x) = \mathcal{D}((\hat{T}_P \Phi)(x)),$$

where $\Phi(y) \sim \psi(y)$ independently of $\hat{T}_P$, and $\mathcal{D}$ extracts the distribution of the input random variable. The purely distributional form of the SC-CDF Bellman equation is then written as a fixed point condition on $\mathscr{P}(\mathscr{F})^{\mathcal{X}}$:

$$\psi = \mathcal{T}_{\text{SCC}} \, \psi. \tag{19}$$

We also write

$$\overline{w}_{\|\cdot\|_{\ell_2}}(\psi, \psi') = \max_{x \in \mathcal{X}} w_{\|\cdot\|_{\ell_2}}(\psi(x), \psi(x'))$$

for the supremum-Wasserstein distance over $\mathscr{P}(\mathscr{F})^{\mathcal{X}}$ with base metric $\|\cdot\|_{\ell_2}$ on $\mathscr{F}$.

**Proposition 5.7.** *The SC-CDF Bellman equation in Equation* (16) *has a unique solution, in the sense that there is a unique distribution for each $\Phi(x)$ such that Equation* (16) *holds for each $x \in \mathcal{X}$.*

*Proof.* We first show that the operator $\mathcal{T}_{\text{SCC}}$ is a contraction on $\mathscr{P}(\mathscr{F})^{\mathcal{X}}$ with respect to the metric $\overline{w}_{\|\cdot\|_{\ell_2}}$. Suppose $\psi, \psi' \in \mathscr{P}(\mathscr{F})^{\mathcal{X}}$, and let $(\Phi(y), \Phi'(y))$ be an optimal coupling between $\psi(y)$ and $\psi'(y)$ with respect to $w_{\|\cdot\|_{\ell_2}}$ for each $y \in \mathcal{X}$ (existence of such couplings is guaranteed by Villani (2009, Theorem 4.1)). Then, letting $(x, X')$ be a random transition from $x$, independent of $\Phi, \Phi'$, we have that $(B_x \Phi(X'), B_x \Phi'(X'))$ is a valid coupling of $(\mathcal{T}_{\text{SCC}} \, \psi)(x)$ and $(\mathcal{T}_{\text{SCC}} \, \psi')(x)$. Then we have, using the operator notation defined in Appendix B,

$$
\begin{aligned}
w_{\|\cdot\|_{\ell_2}}((\mathcal{T}_{\text{SCC}} \, \psi)(x), (\mathcal{T}_{\text{SCC}} \, \psi')(x)) &\overset{(a)}{\leq} \mathbb{E}\left[\|B_x \Phi(X') - B_x \Phi'(X')\|_{\ell_2}\right] \\
&\overset{(b)}{\leq} \sqrt{\gamma}\mathbb{E}\left[\|\Phi(X') - \Phi'(X')\|_{\ell_2}\right] \\
&= \sqrt{\gamma}\sum_{y \in \mathcal{X}} P(y|x)\mathbb{E}\left[\|\Phi(y) - \Phi'(y)\|_{\ell_2}\right] \\
&\overset{(c)}{=} \sqrt{\gamma}\sum_{y \in \mathcal{X}} P(y|x)w_{\|\cdot\|_{\ell_2}}(\psi(y), \psi'(y)) \\
&\leq \sqrt{\gamma}\overline{w}_{\|\cdot\|_{\ell_2}}(\psi, \psi'),
\end{aligned}
$$

where (a) follows since $(B_x\Phi(X'), B_x\Phi'(X'))$ is a valid coupling of $(\mathcal{T}_{\mathrm{SCC}}\,\psi)(x)$ and $(\mathcal{T}_{\mathrm{SCC}}\,\psi')(x)$, (b) follows by contractivity of $B_x$ with respect to $\|\cdot\|_{\ell_2}$, as shown in Proposition B.2, and (c) follows since $(\Phi(y), \Phi'(y))$ was chosen to be an optimal coupling of $\psi(y)$ and $\psi(y')$. Hence we have

$$\overline{w}_{\|\cdot\|_{\ell_2}}(\mathcal{T}_{\mathrm{SCC}}\psi, \mathcal{T}_{\mathrm{SCC}}\psi') \leq \sqrt{\gamma}\,\overline{w}_{\|\cdot\|_{\ell_2}}(\psi, \psi')\,,$$

proving contractivity.

The metric space $(\mathscr{P}(\mathscr{F})^{\mathcal{X}}, \overline{w}_{\|\cdot\|_{\ell_2}})$ is complete, since the base space $(\mathscr{F}, \|\cdot\|_{\ell_2})$ is separable and complete (Villani, 2009, Theorem 6.18). Hence, by Banach's fixed point theorem, we obtain that there is a unique fixed point $\psi^* \in \mathscr{P}(\mathscr{F})^{\mathcal{X}}$ of $\mathcal{T}_{\mathrm{SCC}}$. Thus, a collection of random CDFs $(\Phi^*(x) : x \in \mathcal{X})$ satisfy the SC-CDF Bellman equation in Equation (16) if and only if we have $\Phi^*(x) \sim \psi^*(x)$ for all $x \in \mathcal{X}$. $\qquad\square$

**Proposition 5.8.** *For all $x \in \mathcal{X}$, we have*

$$\mathbb{E}[\Phi^Q(x)] = F^Q(x)\,.$$

*Proof.* We take expectations on both sides of the random-variable stochastic categorical CDF Bellman equation in Equation (16), yielding:

$$\mathbb{E}[\Phi^Q(x)] = \mathbb{E}[(\hat{T}_Q\Phi^Q)(x)]\,.$$

Since $\hat{T}_Q$ is a random linear map, independent of $\Phi^Q$, we have

$$\begin{aligned}\mathbb{E}[\Phi^Q(x)] &= (\mathbb{E}[\hat{T}_Q]\mathbb{E}[\Phi^Q])(x)\\ &= T_Q\mathbb{E}[\Phi^Q](x)\,.\end{aligned}$$

This states that $\mathbb{E}[\Phi^Q] \in \mathbb{R}^{\mathcal{X} \times m}$ satisfies the standard categorical Bellman equation in Equation (10), and hence $\mathbb{E}[\Phi^Q] = F^Q$, by Proposition 2.2, as required. $\qquad\square$

**Proposition 5.11.** *We have*

$$\Sigma_Q \geq \sigma_Q + \gamma Q\Sigma_Q - \left(\frac{2}{m\sqrt{1-\gamma}} + \frac{1}{m^2(1-\gamma)^2}\right)\mathbf{1}\,,$$

*where $\mathbf{1} \in \mathbb{R}^{\mathcal{X}}$ is a vector of ones, and the inequality above is interpreted component-wise.*

*Proof.* We calculate, writing $\Phi^Q$ as $\Phi$, $F^Q$ as $F$, and $\hat{T}_Q$ as $\hat{T}$ to lighten notation:

$$\begin{aligned}\Sigma_Q(x) &= \mathbb{E}[\ell_2^2(\Phi(x), F(x))]\\ &\overset{(a)}{=} \mathbb{E}[\ell_2^2((\hat{T}\Phi)(x), F(x))]\\ &\overset{(b)}{=} \mathbb{E}[\ell_2^2((\hat{T}F)(x), F(x))] + \mathbb{E}[\ell_2^2((\hat{T}\Phi)(x), (\hat{T}F)(x))]\,.\end{aligned} \qquad (20)$$

Here, (a) follows since $\Phi$ satisfies the random-variable version of the stochastic categorical CDF Bellman equation, (b) is a result of a Pythagorean identity for squared Cramér distance, which we derive below:

$$\begin{aligned}\mathbb{E}[\ell_2^2((\hat{T}\Phi)(x), F(x))] &= \mathbb{E}\left[\int_0^{(1-\gamma)^{-1}} ((\hat{T}\Phi)(x)(t) - F(x)(t))^2\,\mathrm{d}t\right]\\ &= \int_0^{(1-\gamma)^{-1}} \mathbb{E}[((\hat{T}\Phi)(x)(t) - F(x)(t))^2]\,\mathrm{d}t\,,\end{aligned}$$

where the integral switch follows from Fubini's theorem. Now, focusing on the integrand above, it can be written as

$$\mathbb{E}[(Y - \mathbb{E}[Y])^2]$$

with $Y = (\hat{T}\Phi)(x)(t)$, since $F(x) = \mathbb{E}[(\hat{T}\Phi)(x)] = \mathbb{E}[\Phi(x)]$, by Lemma 5.8. We have

$$\begin{aligned}\mathbb{E}[(Y - \mathbb{E}[Y])^2] &= \mathbb{E}[(Y - \mathbb{E}[Y|\hat{T}] + \mathbb{E}[Y|\hat{T}] - \mathbb{E}[Y])^2]\\ &= \mathbb{E}[(Y - \mathbb{E}[Y|\hat{T}])^2] + \mathbb{E}[(\mathbb{E}[Y|\hat{T}] - \mathbb{E}[Y])^2]\,.\end{aligned}$$

Now, $\mathbb{E}[(\hat{T}\Phi)(x)(t) \mid \hat{T}] = (\hat{T}\mathbb{E}[\Phi])(x)(t) = (\hat{T}F)(x)(t)$, by linearity, which concludes the validation of step (b) above. We recognise the first term in Equation (20) as the local squared-Cramér variation, and hence have

$$\Sigma_Q(x) = \sigma_Q(x) + \mathbb{E}[\ell_2^2((\hat{T}\Phi)(x), (\hat{T}F)(x))].$$

We now explicitly write the evaluation of the application of $\hat{T}$ at coordinate $x$ in terms of the random transition $(x, X')$ used to construct the single-sample random transition matrix $\hat{Q}$ described in Definition 5.4, so that we obtain, with the operator notation of Appendix B,

$$
\begin{aligned}
\mathbb{E}[\ell_2^2((\hat{T}\Phi)(x), (\hat{T}F)(x))] &= \mathbb{E}[\ell_2^2(B_x\Phi(X'), B_x F(X'))] \\
&= \mathbb{E}[\mathbb{E}[\ell_2^2(B_x\Phi(X'), B_x F(X')) \mid \Phi]] \\
&= \mathbb{E}\left[\sum_{y\in\mathcal{X}} Q(y|x)\ell_2^2(B_x\Phi(y), B_x F(y))\right] \\
&\overset{(a)}{\geq} \sum_{y\in\mathcal{X}} Q(y|x)\mathbb{E}[\gamma\ell_2^2(\Phi(y), F(y)) - \alpha] \\
&= \gamma\sum_{y\in\mathcal{X}} Q(y|x)\mathbb{E}[\ell_2^2(\Phi(y), F(y))] - \alpha \\
&= \gamma\sum_{y\in\mathcal{X}} Q(y|x)\mathbb{E}[\ell_2^2(\Phi(y), F(y))] - \alpha \\
&= \gamma(Q\Sigma_Q)(x) - \alpha,
\end{aligned}
$$

where (a) follows from Proposition B.3, with $\alpha = \frac{2}{m(1-\gamma)^{1/2}} + \frac{1}{m^2(1-\gamma)^2}$, meaning that we deduce

$$\Sigma_Q(x) = \sigma_Q(x) + \gamma(Q\Sigma_Q)(x) - \alpha,$$

as required. $\qquad\square$

**Corollary 5.12.** *We can bound the term $\|(I - \gamma\hat{P})^{-1}\sigma_{\hat{P}}\|_\infty$ appearing in Equation (14) under the assumptions of Theorem 5.3 as follows:*

$$\|(I - \gamma\hat{P})^{-1}\sigma_{\hat{P}}\|_\infty \leq \|\Sigma_{\hat{P}}\|_\infty + \frac{1}{1-\gamma} \leq \frac{2}{1-\gamma}.$$

*Proof.* By Proposition 5.11 applied with $Q = \hat{P}$,

$$\Sigma_{\hat{P}} \geq \sigma_{\hat{P}} + \gamma\hat{P}\Sigma_{\hat{P}} - \left(\frac{2}{m(1-\gamma)^{1/2}} + \frac{1}{m^2(1-\gamma)^2}\right)\mathbf{1},$$

where $\mathbf{1} \in \mathbb{R}^{\mathcal{X}}$ is the vector of ones. We first note that from the condition $m \geq 4\varepsilon^{-2}(1-\gamma)^{-2} + 1$ from the statement of Theorem 5.3, we have

$$\frac{2}{m(1-\gamma)^{1/2}} + \frac{1}{m^2(1-\gamma)^2} \leq \frac{2}{4\varepsilon^{-2}(1-\gamma)^{-2}(1-\gamma)^{1/2}} + \frac{1}{4\varepsilon^{-2}} \leq \frac{\varepsilon^2(1-\gamma)^{3/2}}{2} + \frac{\varepsilon^2}{4} < 1,$$

since $\varepsilon \in (0, 1)$, from the statement of Theorem 5.3.

We therefore have

$$(I - \gamma\hat{P})\Sigma_{\hat{P}} \geq \sigma_{\hat{P}} - \mathbf{1}.$$

Now, $(I - \gamma\hat{P})^{-1}$ is a monotone operator (in the sense that $v_1 \geq v_2$ coordinatewise implies $(I - \gamma\hat{P})^{-1}v_1 \geq (I - \gamma\hat{P})^{-1}v_2$; this claim follows as all elements of $(I - \gamma\hat{P})^{-1}$ are non-negative, since it can also be written $\sum_{k\geq 0}\gamma^k\hat{P}^k$), we can apply it to both sides of the inequality above to obtain

$$\Sigma_{\hat{P}} + (1-\gamma)^{-1}\mathbf{1} \geq (I - \gamma\hat{P})^{-1}\sigma_{\hat{P}}.$$

Finally, we note that since $\Sigma_{\hat{P}}(x)$ is an expected squared-Cramér distance between two CDFs supported on an interval of length $(1-\gamma)^{-1}$, we have

$$\Sigma_{\hat{P}}(x) \leq (1-\gamma)^{-1},$$

from which the claim follows. $\qquad\square$

# E  Proof of Theorem 5.1

We begin by restating the result we seek to prove.

**Theorem 5.1.** *Let $\varepsilon \in (0, (1-\gamma)^{-1/2})$ and $\delta \in (0,1)$, and suppose the number of categories satisfies $m \geq 4(1-\gamma)^{-2}\varepsilon^{-2} + 1$. Then the output $\hat{F}$ of model-based DCFP with $N = \Omega(\varepsilon^{-2}(1-\gamma)^{-3}\mathrm{polylog}(|\mathcal{X}|/\delta))$ samples satisfies, with probability at least $1 - \delta$,*

$$\max_{x \in \mathcal{X}} w_1(\eta^*(x), \hat{F}(x)) \leq \varepsilon \, .$$

We arrange the proof into a sequence of smaller results, in analogy with the presentation of the sketch proof in the main paper.

## E.1  Reduction to high-probability bounds in Cramér distance

Following the sketch provided in the main paper, we first restate and prove Lemma 5.2.

**Lemma 5.2.** *For any two distributions $\nu, \nu' \in \mathscr{P}([0, (1-\gamma)^{-1}])$, we have*

$$w_1(\nu, \nu') \leq (1-\gamma)^{-1/2}\ell_2(\nu, \nu') \, .$$

*Proof.* We begin by writing

$$w_1(\nu, \nu') = \int_0^{(1-\gamma)^{-1}} |F_\nu(t) - F_{\nu'}(t)| \, \mathrm{d}t = (1-\gamma)^{-1}\left[(1-\gamma)\int_0^{(1-\gamma)^{-1}} |F_\nu(t) - F_{\nu'}(t)| \, \mathrm{d}t\right],$$

where $F_\nu, F_{\nu'}$ are the CDFs of $\nu, \nu'$, respectively. The quantity inside the squared brackets can be interpreted as an expectation (with $t$ ranging over the values of a uniform variate on $\mathbb{E}_{T \sim \mathrm{Unif}([0,(1-\gamma)^{-1}])}[|F_\nu(T) - F_{\nu'}(T)|]$, and we can therefore apply Jensen's inequality with the map $z \mapsto z^2$. This then yields

$$\begin{aligned} w_1(\nu, \nu') \leq &(1-\gamma)^{-1}\left[(1-\gamma)\int_0^{(1-\gamma)^{-1}} (F_\nu(t) - F_{\nu'}(t))^2 \, \mathrm{d}t\right]^{1/2} \\ = &(1-\gamma)^{-1/2}\ell_2(\nu, \nu') \, , \end{aligned}$$

as required. $\qquad\square$

The first main step of the proof of Theorem 5.1 is a reduction to Theorem 5.3 via Lemma 5.2. To see this, we first restate Theorem 5.3 here for convenience.

**Theorem 5.3.** *Let $\varepsilon \in (0,1)$ and $\delta \in (0,1)$, and suppose the number of categories satisfies $m \geq 4(1-\gamma)^{-2}\varepsilon^{-2} + 1$. Then the output $\hat{F}$ of model-based DCFP with $N = \Omega(\varepsilon^{-2}(1-\gamma)^{-2}\mathrm{polylog}(|\mathcal{X}|/\delta))$ samples satisfies, with probability at least $1 - \delta$,*

$$\max_{x \in \mathcal{X}} \ell_2(\eta^*(x), \hat{F}(x)) \leq \varepsilon \, . \tag{13}$$

For the proof of the reduction, suppose the statement of Theorem 5.3 holds. Now, let us take $\varepsilon \in (0, (1-\gamma)^{-1/2})$, and $m \geq 4(1-\gamma)^{-2}\varepsilon^{-2} + 1$, as in the assumptions of Theorem 5.1. We then define $\widetilde{\varepsilon} = (1-\gamma)^{1/2}\varepsilon$; note that from the assumption on $\varepsilon$, we have $\widetilde{\varepsilon} \in (0,1)$. Applying the result of Theorem 5.3, we therefore obtain that with $N = \widetilde{\Omega}(\widetilde{\varepsilon}^{-2}(1-\gamma)^{-2}\mathrm{polylog}(|\mathcal{X}|/\delta))$, we have (with probability at least $1 - \delta$)

$$\max_{x \in \mathcal{X}} \ell_2(\eta^*(x), \hat{F}(x)) \leq \widetilde{\varepsilon} \, .$$

By Lemma 5.2, we therefore have (with probability at least $1 - \delta$, for all $x \in \mathcal{X}$)

$$\overline{w}_1(\eta^*(x), \hat{F}(x)) \leq (1-\gamma)^{-1/2}\ell_2(\eta^*(x), \hat{F}(x)) \leq (1-\gamma)^{-1/2}(1-\gamma)^{1/2}\varepsilon = \varepsilon \, ,$$

which is the desired inequality in Theorem 5.1. Finally, we note that the sample complexity term $\widetilde{\varepsilon}^{-2}(1-\gamma)^{-2}$ can be rewritten as

$$\widetilde{\varepsilon}^{-2}(1-\gamma)^{-2} = ((1-\gamma)^{1/2}\varepsilon)^{-2}(1-\gamma)^{-2} = \varepsilon^{-2}(1-\gamma)^{-3} \, .$$

Thus, we obtain the stated sample complexity in Theorem 5.1, and we have established that to prove Theorem 5.1, it is sufficient to prove Theorem 5.3.

## E.2 Reduction to categorical fixed-point error

The first step of the proof of Theorem 5.3 is to show that with $m$ taken sufficiently large (as described in the statement of Theorem 5.3), the Cramér distance between the true return distributions and the categorical fixed points is small, and it is therefore sufficient to focus solely on the sample-based error in estimating the categorical fixed point.

By applying the triangle inequality, we have

$$
\begin{aligned}
\bar{\ell}_2(\eta^*, \hat{F}) &\leq \bar{\ell}_2(\eta^*, F^*) + \bar{\ell}_2(F^*, \hat{F}) \\
&\overset{(a)}{\leq} \frac{1}{(1-\gamma)\sqrt{m-1}} + \bar{\ell}_2(F^*, \hat{F}) \\
&\overset{(b)}{\leq} \frac{\varepsilon}{2} + \bar{\ell}_2(F^*, \hat{F}) \,,
\end{aligned}
$$

where (a) follows from the fixed-point approximation bound in Equation (5), which itself is the result of Proposition 2 of Rowland et al. (2018), and (b) follows from substituting the assumed inequality for $m$ in the statement of Theorem 5.3. Thus, to establish that $\bar{\ell}_2(\eta^*, \hat{F})$ is bounded by $\varepsilon$ with probability at least $1 - \delta$, it suffices to show that

$$
\bar{\ell}_2(F^*, \hat{F}) < \varepsilon/2 \,,
$$

with probability at least $1 - \delta$, as claimed.

## E.3 Propagation of local errors

To begin analysing $\bar{\ell}_2(F^*, \hat{F})$, we analyse the difference of vectors $\hat{F} - F^*$ directly. We proceed in an analogous manner to Azar et al. (2013) in the mean-return case, rearranging as follows:

$$
\begin{aligned}
\hat{F} - F^* &\overset{(a)}{=} T_{\hat{P}}\hat{F} - T_P F^* \\
&\overset{(b)}{=} T_{\hat{P}}\hat{F} - T_{\hat{P}}F^* + T_{\hat{P}}F^* - T_P F^* \\
\implies (I - T_{\hat{P}})(\hat{F} - F^*) &= (T_{\hat{P}} - T_P)F^* \,.
\end{aligned}
\tag{21}
$$

Here, (a) follows since $\hat{F}, F^*$ are fixed points of $T_{\hat{P}}, T_P$, respectively, (b) follows by adding and subtracting $T_{\hat{P}}F^*$, and the implication follows from straightforward rearrangement.

We would next like to rearrange Equation (21) to leave the term $\hat{F} - F^*$ on its own. This requires some care, in checking that the operator $(I - T_{\hat{P}})$ is invertible in an appropriate sense.

**Lemma E.1.** *The operator $I - T_{\hat{P}} : \mathbb{R}^{\mathcal{X} \times m} \to \mathbb{R}^{\mathcal{X} \times m}$ is invertible on the subspace $\{F \in \mathbb{R}^{\mathcal{X} \times m} : F_m(x) = 0 \text{ for all } x \in \mathcal{X}\}$, with inverse $\sum_{k \geq 0} T_{\hat{P}}^k$.*

*Proof.* By Proposition B.1, $T_{\hat{P}}$ maps $\{F \in \mathbb{R}^{\mathcal{X} \times m} : F_m(x) = 0 \text{ for all } x \in \mathcal{X}\}$ to itself, and is a contraction on this subspace with respect to $\ell_2$, with contraction factor $\sqrt{\gamma}$. It therefore follows that $I - T_{\hat{P}}$ maps this subspace to itself, and is invertible on this subspace. Since $\sum_{k \geq 0} T_{\hat{P}}^k$ also maps this subspace to itself, it follows that on this subspace, we have $(I - T_{\hat{P}})^{-1} = \sum_{k \geq 0} T_{\hat{P}}^k$, as required. $\qquad \square$

Now, $(T_{\hat{P}} - T_P)F^* \in \{F \in \mathbb{R}^{\mathcal{X} \times m} : F_m(x) = 0 \text{ for all } x \in \mathcal{X}\}$, and hence from Equation (21) and Lemma E.1, it follows that

$$
\hat{F} - F^* = \sum_{k \geq 0} T_{\hat{P}}^k (T_{\hat{P}} - T_P)F^* \,.
\tag{22}
$$

The result is that we have expressed the difference in CDFs in terms of local errors $(T_{\hat{P}} - T_P)F^*$, which are then propagated via the operator $\sum_{k \geq 0} T_{\hat{P}}^k$.

### E.4 Bernstein concentration bounds

The next step in the proof is to establish a concentration bound for the local errors $(T_{\hat{P}} - T_P)F^*$.

One potential approach is to use a concentration inequality for each individual coordinate of $(T_{\hat{P}} - T_P)F^*$, indexed by $(x, i)$. Azar et al. (2013) note that in the mean-return case, using a Hoeffding-style bound is insufficient to obtain optimal dependence of the sample complexity on $(1 - \gamma)^{-1}$, and Zhang et al. (2023) also note this in their (non-categorical) distributional analysis. Using a Bernstein concentration inequality on each coordinate and then using a union bound can be made to work, although this then incurs a $\log(m)$ factor in the sample complexity. If such a dependence on $m$ were tight, this would suggest that the sample complexity depends on $m$ (albeit only logarithmically), and hence there would be a trade-off between picking $m$ sufficiently large so as to obtain a low representation approximation error, as in Section E.2, and taking $m$ low so as to not unduly increase the sample complexity. However, such a dependence on $m$ can be avoided by working with concentration inequalities at the level of CDFs themselves. The Dvorestsky-Kiefer-Wolfowitz (DKW) inequality (Dvoretzky et al., 1956; Massart, 1990) behaves analogously to Hoeffding's bound, and is insufficient for obtaining a sample complexity bound with optimal $(1 - \gamma)^{-1}$ dependence. Instead, we make use of a Hilbert space Bernstein-style inequality (Yurinsky, 2006; Chatalic et al., 2022), by interpreting the Cramér distance $\ell_2$ as a maximum mean discrepancy (Gretton et al., 2012), which measures distances between distributions via embeddings in a (reproducing kernel) Hilbert space, allowing the Bernstein result to be applied.

First, we precisely describe the connection between Cramér distance and Hilbert space. Székely (2003) shows that for any distributions $\nu, \nu' \in \mathscr{P}([0, (1 - \gamma)^{-1}])$, we have

$$\ell_2^2(\nu, \nu') = \mathbb{E}_{X \sim \nu, Y \sim \nu'}[|X - Y|] - \frac{1}{2}\mathbb{E}_{X, X' \sim \nu}[|X - X'|] - \frac{1}{2}\mathbb{E}_{Y, Y' \sim \nu'}[|Y - Y'|]. \tag{23}$$

Sejdinovic et al. (2013) show that there exists an affine embedding $\phi : \mathscr{P}([0, (1 - \gamma)^{-1}]) \to \mathcal{H}$ of distributions into a Hilbert space $\mathcal{H}$, such that the right-hand side of the equation can also be written as the squared norm $\|\phi(\nu) - \phi(\nu')\|_{\mathcal{H}}^2$; in other words, they show that the Cramér distance is equal to a squared maximum mean discrepancy (Gretton et al., 2012). We can then appeal to the following Bernstein-style inequality for Hilbert-space-valued random variables, which is due to Chatalic et al. (2022), and based largely on Yurinsky (2006, Theorem 3.3.4). We state the result here in the usual Bernstein style of assuming almost-sure boundedness of the the random variables concerned, while Chatalic et al. (2022) use the more general formulation of assuming particular bounds on all moments.

**Lemma E.2.** *(Chatalic et al., 2022, Lemma E.3) Let $Z_1, \ldots, Z_N$ be i.i.d. mean-zero random variables taking values in a Hilbert space $(\mathcal{H}, \|\cdot\|_{\mathcal{H}})$ such that $\|Z_1\|_{\mathcal{H}} \leq M$ almost surely. Then for any $\delta \in (0, 1)$, we have*

$$\left\| \frac{1}{N} \sum_{i=1}^{N} Z_i \right\|_{\mathcal{H}} \leq \frac{2M \log(2/\delta)}{N} + \sqrt{\frac{2\mathbb{E}[\|Z_1\|_{\mathcal{H}}^2] \log(2/\delta)}{N}},$$

*with probability at least $1 - \delta$.*

To apply this result in our setting, first note that, using the notation introduced in Appendix B, we have

$$
\begin{aligned}
\left\| [(T_{\hat{P}} - T_P)F^*](x) \right\|_{\ell_2} &= \ell_2((T_{\hat{P}}F^*)(x), F^*(x)) \\
&= \|\phi((T_{\hat{P}}F^*)(x)) - \phi(F^*(x))\|_{\mathcal{H}} \\
&\overset{(a)}{=} \left\| \phi\left( \frac{1}{N} \sum_{i=1}^{N} F^*(X_i^x)B_x^\top \right) - \phi(F^*(x)) \right\|_{\mathcal{H}} \\
&\overset{(b)}{=} \left\| \frac{1}{N} \sum_{i=1}^{N} \left( \phi\left(F^*(X_i^x)B_x^\top\right) - \phi(F^*(x)) \right) \right\|_{\mathcal{H}},
\end{aligned}
$$

where (a) follows from the expressions for $T_{\hat{P}}$ described in Appendix B, and (b) follows from the affineness of the embedding $\phi$. Here again, we make use of a slight abuse of notation by writing $\phi(F^*(x))$ for the embedding of the distribution supported on $\{z_1, \ldots, z_m\}$, with CDF values $F^*(x)$, and similarly for embeddings of other CDF vectors.

We can now apply the result above, noting that

$$\|\phi\left(F^*(X_i^x)B_x^\top\right) - \phi(F^*(x))\|_{\mathcal{H}} = \ell_2(F^*(X_i^x)B_x^\top, F^*(x)) \le (1-\gamma)^{-1/2}$$

almost surely, and

$$\mathbb{E}\left[\|\phi\left(F^*(X_i^x)B_x^\top\right) - \phi(F^*(x))\|_{\mathcal{H}}^2\right] = \mathbb{E}\left[\ell_2^2(F^*(X_i^x)B_x^\top, F^*(x))\right] = \sigma(x)\,,$$

where we write $\sigma$ as shorthand for the local squared-Cramér variation at $P$, introduced in Definition 5.5 with the notation $\sigma_P$. Hence, from Lemma E.2 and a union bound over the state space $\mathcal{X}$, we obtain that with probability $1 - \delta/2$, we have for all $x \in \mathcal{X}$ that

$$\left\|[(T_{\hat{P}} - T_P)F^*](x)\right\|_{\ell_2} \le C_{\mathrm{B}1}\frac{\sqrt{\sigma(x)}}{\sqrt{N}} + C_{\mathrm{B}2}\frac{1}{(1-\gamma)^{1/2}N}\,, \tag{24}$$

where

$$C_{\mathrm{B}1} = \sqrt{2\log(4|\mathcal{X}|/\delta)}\,, \quad \text{and } C_{\mathrm{B}2} = 2\log(4|\mathcal{X}|/\delta)\,.$$

### E.5 From population to empirical squared-Cramér variation

Next, we show that the population local squared-Cramér variation in Equation (24) can be replaced with the corresponding *empirical* quantity; that is, the local-squared Cramér variation at $\hat{P}$, $\sigma_{\hat{P}}$, incurring some additional terms in the bound. We start by calculating as follows:

$$\sigma(x) = \mathbb{E}\left[\|(\hat{T}_P F^*)(x) - (T_P F^*)(x)\|_{\ell_2}^2\right]$$

$$= \mathbb{E}\left[\|F^*(X')B_x^\top\|_{\ell_2}^2\right] - \|P_x F^* B_x^\top\|_{\ell_2}^2$$

$$= P_x\|F^* B_x^\top\|_{\ell_2}^2 - \|P_x F^* B_x^\top\|_{\ell_2}^2\,,$$

where above we use the notation $\|F^* B_x^\top\|_{\ell_2}^2 \in \mathbb{R}^{\mathcal{X}}$ for the vector indexed by state, so that the element corresponding to state $y \in \mathcal{X}$ is $\|F^*(y)B_x^\top\|_{\ell_2}^2$. To relate this quantity to $\hat{\sigma}(x)$, we add and subtract a term, motivated by the derivation applied to standard return variance by Azar et al. (2013, Lemma 5), and rearrange as follows:

$$\sigma(x)$$
$$= P_x\|F^* B_x^\top\|_{\ell_2}^2 - \|P_x F^* B_x^\top\|_{\ell_2}^2$$
$$= P_x\|F^* B_x^\top\|_{\ell_2}^2 - \|P_x F^* B_x^\top\|_{\ell_2}^2 - \left(\hat{P}_x\|F^* B_x^\top\|_{\ell_2}^2 - \|\hat{P}_x F^* B_x^\top\|_{\ell_2}^2\right)$$
$$\quad + \left(\hat{P}_x\|F^* B_x^\top\|_{\ell_2}^2 - \|\hat{P}_x F^* B_x^\top\|_{\ell_2}^2\right)$$
$$= (P_x - \hat{P}_x)\|F^* B_x^\top\|_{\ell_2}^2 + \left(\|\hat{P}_x F^* B_x^\top\|_{\ell_2}^2 - \|P_x F^* B_x^\top\|_{\ell_2}^2\right)$$
$$\quad + \mathbb{E}\left[\|(\hat{T}_{\hat{P}} F^*)(x) - (T_{\hat{P}} F^*)(x)\|_{\ell_2}^2 \,\Big|\, \hat{P}\right]$$
$$= (P_x - \hat{P}_x)\|F^* B_x^\top\|_{\ell_2}^2 + \langle \hat{P}_x F^* B_x^\top - P_x F^* B_x^\top, \hat{P}_x F^* B_x^\top + P_x F^* B_x^\top\rangle_{\ell_2}$$
$$\quad + \mathbb{E}\left[\|(\hat{T}_{\hat{P}} F^*)(x) - (T_{\hat{P}} F^*)(x)\|_{\ell_2}^2 \,\Big|\, \hat{P}\right]$$
$$= (P_x - \hat{P}_x)\|F^* B_x^\top\|_{\ell_2}^2 + \langle (\hat{P}_x - P_x) F^* B_x^\top, \hat{P}_x F^* B_x^\top + P_x F^* B_x^\top\rangle_{\ell_2}$$
$$\quad + \mathbb{E}\left[\|(\hat{T}_{\hat{P}} F^*)(x) - (T_{\hat{P}} F^*)(x)\|_{\ell_2}^2 \,\Big|\, \hat{P}\right]\,, \tag{25}$$

where $\langle \, , \, \rangle_{\ell_2}$ is the inner product on $\mathbb{R}^m$ corresponding to $\|\cdot\|_{\ell_2}$, defined by

$$\langle F, F'\rangle_{\ell_2} = \frac{1}{m(1-\gamma)}\sum_{i=1}^{m} F_i F_i'\,.$$

Focusing on the final term on the right-hand side of Equation (25), we have

$$\mathbb{E}\left[\|(\hat{T}_{\hat{P}}F^*)(x) - (T_{\hat{P}}F^*)(x)\|_{\ell_2}^2 \,\Big|\, \hat{P}\right]$$

$$=\mathbb{E}\left[\|(\hat{T}_{\hat{P}}F^*)(x) - (\hat{T}_{\hat{P}}\hat{F})(x) + (\hat{T}_{\hat{P}}\hat{F})(x) - (T_{\hat{P}}F^*)(x) + (T_{\hat{P}}\hat{F})(x) - (T_{\hat{P}}\hat{F})(x)\|_{\ell_2}^2 \,\Big|\, \hat{P}\right]$$

$$=\mathbb{E}\left[\|(\hat{T}_{\hat{P}}F^* - \hat{T}_{\hat{P}}\hat{F})(x) - (T_{\hat{P}}F^* - T_{\hat{P}}\hat{F})(x) + (\hat{T}_{\hat{P}}\hat{F})(x) - (T_{\hat{P}}\hat{F})(x)\|_{\ell_2}^2 \,\Big|\, \hat{P}\right]$$

$$=\mathbb{E}\Big[\|(\hat{T}_{\hat{P}}F^* - \hat{T}_{\hat{P}}\hat{F})(x) - (T_{\hat{P}}F^* - T_{\hat{P}}\hat{F})(x)\|_{\ell_2}^2$$
$$\qquad + 2\langle(\hat{T}_{\hat{P}}F^* - \hat{T}_{\hat{P}}\hat{F})(x) - (T_{\hat{P}}F^* - T_{\hat{P}}\hat{F})(x), (\hat{T}_{\hat{P}}\hat{F} - T_{\hat{P}}\hat{F})(x)\rangle_{\ell_2}$$
$$\qquad\quad + \|(\hat{T}_{\hat{P}}\hat{F})(x) - (T_{\hat{P}}\hat{F})(x)\|_{\ell_2}^2 \,\Big|\, \hat{P}\Big]$$

$$\overset{(a)}{\leq} \bigg(\mathbb{E}[\|(\hat{T}_{\hat{P}}F^* - \hat{T}_{\hat{P}}\hat{F})(x) - (T_{\hat{P}}F^* - T_{\hat{P}}\hat{F})(x)\|_{\ell_2}^2 \mid \hat{P}]^{1/2}$$

$$\qquad + \mathbb{E}[\|(\hat{T}_{\hat{P}}\hat{F})(X) - (T_{\hat{P}}\hat{F})(x)\|_{\ell_2}^2 \mid \hat{P}]^{1/2}\bigg)^2$$

$$= \bigg(\mathbb{E}[\|(\hat{T}_{\hat{P}}F^* - \hat{T}_{\hat{P}}\hat{F})(x) - (T_{\hat{P}}F^* - T_{\hat{P}}\hat{F})(x)\|_{\ell_2}^2 \mid \hat{P}]^{1/2} + \sqrt{\hat{\sigma}(x)}\bigg)^2.$$

where (a) follows from the Cauchy-Schwarz inequality, and we write $\hat{\sigma}$ as a shorthand for $\sigma_{\hat{P}}$. Finally, we note that

$$\mathbb{E}[\|(\hat{T}_{\hat{P}}F^* - \hat{T}_{\hat{P}}\hat{F})(x) - (T_{\hat{P}}F^* - T_{\hat{P}}\hat{F})(x)\|_{\ell_2}^2 \mid \hat{P}] \leq \mathbb{E}[\|(\hat{T}_{\hat{P}}F^* - \hat{T}_{\hat{P}}\hat{F})(x)\|_{\ell_2}^2 \mid \hat{P}]$$

$$\overset{(a)}{\leq} \gamma\mathbb{E}_{X'\sim\hat{P}_x}[\|F^*(X') - \hat{F}(X')\|_{\ell_2}^2 \mid \hat{P}]$$

$$\leq \gamma\|F^* - \hat{F}\|_{\ell_2,\infty}^2\,,$$

with (a) following from contractivity of $B_x$ in $\|\cdot\|_{\ell_2}$, as established in Proposition B.2. Bringing these bounds together with Equation (25), we have

$$\sigma(x) \leq (P_x - \hat{P}_x)\|F^* B_x^\top\|_{\ell_2}^2 + \langle(\hat{P}_x - P_x)F^* B_x^\top, \hat{P}_x F^* B_x^\top + P_x F^* B_x^\top\rangle_{\ell_2} \qquad (26)$$

$$+ \left(\gamma\|F^* - \hat{F}\|_{\ell_2,\infty} + \sqrt{\hat{\sigma}(x)}\right)^2.$$

We now apply concentration bounds to each of the first two terms on the right-hand side of Equation (26).

**Lemma E.3.** *With probability at least $1 - \delta/2$, we have for all $x \in \mathcal{X}$,*

$$\left|P_x\|F^* B_x^\top\|_{\ell_2}^2 - \hat{P}_x\|F^* B_x^\top\|_{\ell_2}^2\right| \leq C_H \frac{1}{(1-\gamma)\sqrt{N}}\,,$$

*where*

$$C_H = \sqrt{\frac{\log(4|\mathcal{X}|/\delta)}{2}}\,.$$

*Proof.* The expression $P_x\|F^* B_x^\top\|_{\ell_2}^2 - \hat{P}_x\|F^* B_x^\top\|_{\ell_2}^2$ is equal to the negative of the average of $N$ i.i.d. copies of the random variable $\|F^*(X')B_x^\top\|_{\ell_2}^2$, where $X' \sim P_x$, minus its expectation. Since $\|F^*(X')B_x^\top\|_{\ell_2}^2$ is bounded in $[0, (1-\gamma)^{-1}]$, we apply Hoeffding's inequality (for a given $x \in \mathcal{X}$) to obtain

$$\left|P_x\|F^* B_x^\top\|_{\ell_2}^2 - \hat{P}_x\|F^* B_x^\top\|_{\ell_2}^2\right| \leq \frac{1}{(1-\gamma)\sqrt{N}}\sqrt{\frac{\log(4|\mathcal{X}|/\delta)}{2}}\,,$$

with probability at least $1 - \delta/(2|\mathcal{X}|)$. Taking a union bound over $x \in \mathcal{X}$ then yields the statement. $\square$

For the second term, we may re-use the Bernstein concentration bound derived above to deduce the following.

**Lemma E.4.** *Suppose Equation* (24) *holds. Then we have, for all $x \in \mathcal{X}$,*

$$\left| \langle (\hat{P}_x - P_x) F^* B_x^\top, (\hat{P}_x + P_x) F^* B_x^\top \rangle_{\ell_2} \right| \leq 2 C_{\mathrm{B}1} \frac{1}{(1 - \gamma)\sqrt{N}} + 2 C_{\mathrm{B}2} \frac{1}{(1 - \gamma)N} .$$

*Proof.* First, by the Cauchy-Schwarz inequality, we have

$$\left| \langle (\hat{P}_x - P_x) F^* B_x^\top, (\hat{P}_x + P_x) F^* B_x^\top \rangle_{\ell_2} \right| \leq \| (\hat{P}_x - P_x) F^* B_x^\top \|_{\ell_2} \| (\hat{P}_x + P_x) F^* B_x^\top \|_{\ell_2} .$$

The first of the two factors in the product on the right-hand side is precisely the term bounded on the left-hand side of Equation (24). On the right-hand side, the vector inside the $\ell_2$-norm has components in $[0, 2]$, so the norm can be straightforwardly bounded by

$$\left[ \frac{1}{m(1 - \gamma)} \sum_{i=1}^m 2^2 \right]^{1/2} = \frac{2}{(1 - \gamma)^{1/2}} .$$

Combining the inequalities for these two factors, and using the trivial bound $\sqrt{\sigma(x)} \leq (1 - \gamma)^{-1/2}$, we obtain the stated result. $\square$

Combining all these bounds together in Equation (26), and taking a union bound, we conclude that with probability at least $1 - \delta/2$, for all $x \in \mathcal{X}$ we have

$$\sigma(x) \leq (2 C_{\mathrm{B}1} + C_{\mathrm{H}}) \frac{1}{(1 - \gamma)\sqrt{N}} + 2 C_{\mathrm{B}2} \frac{1}{(1 - \gamma)N} + \left( \gamma \| F^* - \hat{F} \|_{\ell_2,\infty} + \sqrt{\hat{\sigma}(x)} \right)^2 ,$$

We now take square-roots of both sides, using the inequality $\sqrt{a + b} \leq \sqrt{a} + \sqrt{b}$ on the right-hand side to obtain

$$\sqrt{\sigma(x)} \leq \sqrt{\hat{\sigma}(x)} + \gamma \| F^* - \hat{F} \|_{\ell_2,\infty} + \sqrt{2 C_{\mathrm{B}1} + C_{\mathrm{H}}} \frac{1}{(1 - \gamma)^{1/2} N^{1/4}} + \sqrt{2 C_{\mathrm{B}2}} \frac{1}{(1 - \gamma)^{1/2} N^{1/2}} .$$

Substituting this into Equation (24) and taking a union bound then yields that with probability at least $1 - \delta$, we have

$$\left\| [(T_{\hat{P}} - T_P) F^*](x) \right\|_{\ell_2} \leq C_{\mathrm{B}1} \frac{1}{\sqrt{N}} \sqrt{\hat{\sigma}(x)} + C_{\mathrm{B}1} \frac{1}{\sqrt{N}} \| F^* - \hat{F} \|_{\ell_2,\infty} + C' \frac{1}{(1 - \gamma)^{1/2} N^{3/4}} , \tag{27}$$

where

$$C' = C_{\mathrm{B}1} (\sqrt{2 C_{\mathrm{B}1} + C_{\mathrm{H}}} + \sqrt{2 C_{\mathrm{B}2}}) + C_{\mathrm{B}2} ,$$

so that our concentration inequality is expressed in terms of the empirical local squared-Cramér variation $\hat{\sigma}$.

### E.6 Converting local bounds to global bounds

To convert the local concentration results obtained above into a bound on $\ell_2(\hat{F}(x), F^*(x))$, we first prove the following lemma.

**Lemma E.5.** *For $U \in \{ F \in \mathbb{R}^{\mathcal{X} \times m} : F_m(x) = 0 \text{ for all } x \in \mathcal{X} \}$, and any $k \geq 1$, we have*

$$\| (T_{\hat{P}}^k U)(x) \|_{\ell_2} \leq \gamma^{k/2} \sum_{x' \in \mathcal{X}} \mathbb{P}_{\hat{P}}(X_k = x' \mid X_0 = x) \| U(x') \|_{\ell_2} ,$$

*where $\mathbb{P}_{\hat{P}}$ denotes state transition probabilities for the MRP with transition matrix $\hat{P}$.*

*Proof.* We proceed by induction. For the base case $k = 1$, we have, using the operator notation introduced in Appendix B,

$$\|(T_{\hat{P}}U)(x)\|_{\ell_2} = \left\|\sum_{y \in \mathcal{X}} \hat{P}(y|x) \sum_{j=1}^{m-1} B_x U(y)\right\|_{\ell_2}$$

$$\overset{(a)}{\leq} \sum_{y \in \mathcal{X}} \hat{P}(y|x) \left\| B_x U(y)\right\|_{\ell_2}$$

$$\overset{(b)}{\leq} \sum_{y \in \mathcal{X}} \hat{P}(y|x) \gamma^{1/2} \left\| U(y)\right\|_{\ell_2},$$

as required. where (a) follows from the triangle inequality, and (b) follows from contractivity of $B_x$ in $\|\cdot\|_{\ell_2}$, as established by Proposition B.2.

For the inductive step, we suppose that the result holds for some $l \in \mathbb{N}$. Now, letting $k = l+1$, we have

$$\|(T_{\hat{P}}^{l+1}U)(x)\|_{\ell_2} = \|(T_{\hat{P}}T_{\hat{P}}^l U)(x)\|_{\ell_2}$$

$$= \left\|\sum_{y \in \mathcal{X}} \hat{P}(y|x) B_x (T_{\hat{P}}^l U)(y)\right\|_{\ell_2}$$

$$\leq \sum_{y \in \mathcal{X}} \hat{P}(y|x) \left\| B_x (T_{\hat{P}}^l U)(y)\right\|_{\ell_2}$$

$$\leq \sum_{y \in \mathcal{X}} \hat{P}(y|x) \gamma^{1/2} \left\| (T_{\hat{P}}^l U)(y)\right\|_{\ell_2}$$

$$\overset{(a)}{\leq} \sum_{y \in \mathcal{X}} \hat{P}(y|x) \gamma^{1/2} \gamma^{l/2} \sum_{x' \in \mathcal{X}} \mathbb{P}_{\hat{P}}(X_l = x' \mid X_0 = y) \|U(x')\|_{\ell_2}$$

$$= \gamma^{(l+1)/2} \sum_{x' \in \mathcal{X}} \left[\sum_{y \in \mathcal{X}} \mathbb{P}_{\hat{P}}(X_l = x' \mid X_0 = y) \hat{P}(y|x)\right] \|U(x')\|_{\ell_2}$$

$$= \gamma^{(l+1)/2} \sum_{x' \in \mathcal{X}} \mathbb{P}_{\hat{P}}(X_{l+1} = x' \mid X_0 = x) \|U(x')\|_{\ell_2},$$

as required, where (a) follows by the inductive hypothesis. $\qquad \square$

We therefore have, with probability at least $1 - \delta$:

$$\ell_2(\hat{F}(x), F^*(x)) \tag{28}$$

$$= \|\hat{F}(x) - F^*(x)\|_{\ell_2}$$

$$\overset{(a)}{=} \left\|\left[\sum_{k \geq 0} T_{\hat{P}}^k (T_{\hat{P}} - T_P) F^*\right](x)\right\|_{\ell_2}$$

$$\overset{(b)}{\leq} \sum_{k \geq 0} \left\|\left[T_{\hat{P}}^k (T_{\hat{P}} - T_P) F^*\right](x)\right\|_{\ell_2}$$

$$\overset{(c)}{\leq} \sum_{k \geq 0} \sum_{x' \in \mathcal{X}} \mathbb{P}_{\hat{P}}(X_k = x' \mid X_0 = x) \gamma^{k/2} \left\|\left[(T_{\hat{P}} - T_P) F^*\right](x')\right\|_{\ell_2}$$

$$\overset{(d)}{\leq} \sum_{k \geq 0} \sum_{x' \in \mathcal{X}} \mathbb{P}_{\hat{P}}(X_k = x' \mid X_0 = x) \gamma^{k/2} \left[C_{\text{B1}} \frac{1}{\sqrt{N}} \sqrt{\hat{\sigma}(x')} + C_{\text{B1}} \frac{1}{\sqrt{N}} \|F^* - \hat{F}\|_{\ell_2, \infty}\right.$$

$$\left. + C' \frac{1}{(1 - \gamma)^{1/2} N^{3/4}}\right]$$

$$\overset{(e)}{\leq} \frac{C_{\text{B1}}}{\sqrt{N}} \|(I - \sqrt{\gamma}\hat{P})^{-1}\sqrt{\hat{\sigma}}\|_\infty + \frac{C_{\text{B1}}}{(1 - \sqrt{\gamma})\sqrt{N}} \|F^* - \hat{F}\|_{\ell_2, \infty} + \frac{C'}{(1 - \sqrt{\gamma})(1 - \gamma)^{1/2} N^{3/4}}$$

$$\overset{(f)}{\leq} \frac{C_{\text{B1}}}{\sqrt{N}} \|(I - \sqrt{\gamma}\hat{P})^{-1}\sqrt{\hat{\sigma}}\|_\infty + \frac{2C_{\text{B1}}}{(1 - \gamma)\sqrt{N}} \|F^* - \hat{F}\|_{\ell_2, \infty} + \frac{2C'}{(1 - \gamma)^{3/2} N^{3/4}} . \tag{29}$$

Here, (a) follows from Equation (22), (b) follows from the triangle inequality, and (c) follows from Lemma E.5. The step (d) is the one step in the derivation that does not hold with probability 1, but rather with probability $1 - \delta$, and follows from the Bernstein-style bounds in Equation (27), (e) follows from algebraic manipulation using the identity $\sum_{k \geq 0}(\gamma^{1/2}\hat{P})^k = (I - \sqrt{\gamma}\hat{P})^{-1}$, as $\hat{P}$ is a stochastic matrix, and bounding the elements of $(I - \sqrt{\gamma}\hat{P})^{-1}\sqrt{\hat{\sigma}}$ by the $L^\infty$ norm of the vector, and (f) follows from the inequality $(1 - \sqrt{\gamma})^{-1} \leq 2(1 - \gamma)^{-1}$ for $\gamma \in [0, 1)$.

Focusing now on the coefficient $\|(I - \sqrt{\gamma}\hat{P})^{-1}\sqrt{\hat{\sigma}}\|_\infty$, we note that similar to the analysis of Agarwal et al. (2020, Lemma 4) in the mean-return case, we can write

$$(I - \sqrt{\gamma}\hat{P})^{-1}\sqrt{\hat{\sigma}} = (1 - \sqrt{\gamma})^{-1}(1 - \sqrt{\gamma})(I - \sqrt{\gamma}\hat{P})^{-1}\sqrt{\hat{\sigma}} ,$$

so that each row of

$$(1 - \sqrt{\gamma})(I - \sqrt{\gamma}\hat{P})^{-1}$$

forms a probability distribution, and so we may apply Jensen's inequality with the map $z \mapsto \sqrt{z}$, to obtain

$$(I - \sqrt{\gamma}\hat{P})^{-1}\sqrt{\hat{\sigma}} \leq (1 - \sqrt{\gamma})^{-1}\sqrt{(1 - \sqrt{\gamma})(I - \sqrt{\gamma}\hat{P})^{-1}\hat{\sigma}} = (1 - \sqrt{\gamma})^{-1/2}\sqrt{(I - \sqrt{\gamma}\hat{P})^{-1}\hat{\sigma}} , \tag{30}$$

where the inequality above holds coordinate-wise. We thus obtain the inequality

$$\|F^* - \hat{F}\|_{\ell_2, \infty} \leq$$

$$\frac{2C_{\text{B1}}}{(1 - \gamma)^{1/2}\sqrt{N}} \sqrt{\|(I - \sqrt{\gamma}\hat{P})^{-1}\sqrt{\hat{\sigma}}\|_\infty} + \frac{2C_{\text{B1}}}{(1 - \gamma)\sqrt{N}} \|F^* - \hat{F}\|_{\ell_2, \infty} + \frac{2C'}{(1 - \gamma)^{3/2} N^{3/4}} .$$

Note that the term $\|F^* - \hat{F}\|_{\ell_2, \infty}$ appears on both sides of the inequality above. By taking $N \geq 16C_{\text{B1}}^2(1 - \gamma)^{-2}$, we ensure that the coefficient in front of $\|F^* - \hat{F}\|_{\ell_2, \infty}$ on the right-hand side is made smaller than $1/2$, similar to the approach taken in the proof of instance-dependent sample complexity bounds for mean-return estimation by Pananjady and Wainwright (2020, Theorem 1(a)). Under this assumption, rearrangement yields

$$\|F^* - \hat{F}\|_{\ell_2, \infty} \leq \frac{4C_{\text{B1}}}{(1 - \gamma)^{1/2}\sqrt{N}} \sqrt{\|(I - \sqrt{\gamma}\hat{P})^{-1}\sqrt{\hat{\sigma}}\|_\infty} + \frac{4C'}{(1 - \gamma)^{3/2} N^{3/4}} \tag{31}$$

as described in Section 5.

### E.7 The stochastic categorical CDF Bellman equation

We now aim to use our developments regarding the stochastic categorical CDF Bellman equation (see Section 5.2) to bound the quantity appearing within the square root. To be able to use the bound obtained in Corollary 5.12, we first replace the factor $\sqrt{\gamma}$ inside the square-root on the right-hand side of Equation (30) with $\gamma$, using the bound $(I - \sqrt{\gamma}P)^{-1} \leq 2(I - \gamma P)^{-1}$, shown by Agarwal et al. (2020, Lemma 4). This, together with Corollary 5.12, yields

$$\sqrt{(I - \sqrt{\gamma}\hat{P})^{-1}\hat{\sigma}} \leq \sqrt{2}\sqrt{(I - \gamma\hat{P})^{-1}\hat{\sigma}} \leq \sqrt{2}\sqrt{\frac{2}{1-\gamma}}\mathbf{1} = \frac{2}{(1-\gamma)^{1/2}}\mathbf{1}\,.$$

Thus, returning to Equation (31) and substituting this bound in, we obtain (again, with probability at least $1 - \delta$, for all $x \in \mathcal{X}$):

$$\ell_2(\hat{F}(x), F^*(x)) \leq \frac{8C_{\mathrm{B}1}}{(1-\gamma)\sqrt{N}} + \frac{4C'}{(1-\gamma)^{3/2}N^{3/4}}\,. \tag{32}$$

### E.8 Final steps of the proof of Theorem 5.3

We now let $N \geq c_0\varepsilon^{-2}(1-\gamma)^{-2}$, with $c_0$ any positive number satisfying

$$c_0 \geq 2^{10}C_{\mathrm{B}1}^2 \text{ and } c_0 > 16^{4/3}(C')^{4/3}\,. \tag{33}$$

Note that the first of these conditions implies the earlier assumption $N \geq 16C_{\mathrm{B}1}^2(1-\gamma)^{-2}$ made in order to arrive at Equation (31). We then conclude from Equation (32) that (with probability at least $1 - \delta$, for all $x \in \mathcal{X}$):

$$\begin{aligned}
\ell_2(\hat{F}(x), F^*(x)) &\leq \frac{8C_{\mathrm{B}1}}{\sqrt{c_0}}\varepsilon + \frac{4C'}{c_0^{3/4}}\varepsilon^{3/2} \\
&\leq \frac{8C_{\mathrm{B}1}}{\sqrt{c_0}}\varepsilon + \frac{4C'}{c_0^{3/4}}\varepsilon \\
&\leq \varepsilon/2\,,
\end{aligned}$$

with the final inequality following due to the assumption on $c_0$ making both coefficients of $\varepsilon$ bounded by $1/4$. This concludes the proof of Theorem 5.3.

To derive a concrete sample complexity bound, including logarithmic terms, we may bound the terms in Equation (33) as follows; we emphasise that we do not aim to be as tight as possible in the following analysis, but merely to provide an concrete, valid bound on $c_0$. First, we have that the

$$\begin{aligned}
2^{10}C_{\mathrm{B}1}^2 &= 2^{10}(\sqrt{2\log(4|\mathcal{X}|/\delta)})^2 \\
&= 2^{11}\log(4|\mathcal{X}|/\delta)\,.
\end{aligned}$$

Next, we have (introducing the shorthand $L = \log(4|\mathcal{X}|/\delta)$):

$$\begin{aligned}
16^{4/3}(C')^{4/3} &\leq 2^6\left(C_{\mathrm{B}1}(\sqrt{2C_{\mathrm{B}1} + C_{\mathrm{H}}} + \sqrt{2C_{\mathrm{B}2}}) + C_{\mathrm{B}2}\right)^{4/3} \\
&= 2^6\left(\sqrt{2L}(\sqrt{2\sqrt{2L} + \sqrt{L/2}} + \sqrt{4L}) + 2L\right)^{4/3} \\
&\leq 2^6\left(L\sqrt{2}(\sqrt{2\sqrt{2} + \sqrt{1/2}} + 2) + 2L\right)^{4/3} \\
&= 2^6L^{4/3}\left(\sqrt{2}(\sqrt{2\sqrt{2} + \sqrt{1/2}} + 2) + 2\right)^{4/3} \\
&\leq 2^6L^{4/3}2^4 \\
&= 2^{10}L^{4/3}\,.
\end{aligned}$$

We therefore conclude, for example, that taking $c_0 \geq 2^{11}\log(4|X|/\delta)^{4/3}$ is sufficient.

## F  Extensions

Below, we describe several straightforward extensions of Theorem 5.1, that allow us to give guarantees for related algorithms and problems concerning sample-efficient distributional reinforcement learning.

## F.1 Categorical dynamic programming

The core theoretical result, Theorem 5.1, can be used to provide guarantees for the iterative CDP algorithm. Let us write $F_k \in \mathbb{R}^{\mathcal{X} \times [m]}$ for the result of applying the empirical CDP operator $\Pi_m \mathcal{T}$ a total $k$ times to an initial estimate $F_0 \in \mathbb{R}^{\mathcal{X} \times [m]}$. By the existing categorical theory described in Proposition 2.2 (Rowland et al., 2018), we can bound the distance between the CDP estimate $F_k$ and the DCFP solution $\hat{F}$ as

$$\overline{\ell}_2(F_k, \hat{F}) \leq \gamma^{k/2} \overline{\ell}_2(F_0, \hat{F}) \leq \frac{\gamma^{k/2}}{(1-\gamma)^{1/2}} \,.$$

Thus, taking $k \geq \frac{2 \log(1/\varepsilon) + 3 \log(1/(1-\gamma))}{\log(1/\gamma)}$, this error is smaller than $\varepsilon(1-\gamma)^{1/2}$. Thus, by Lemma 5.2, we have

$$\overline{w}_1(F_k, \hat{F}) \leq \varepsilon \,.$$

By the triangle inequality, we therefore have that if $\overline{w}_1(\hat{F}, \eta^*) \leq \varepsilon$, then $\overline{w}_1(F_k, \eta^*) \leq 2\varepsilon$ under this assumption on $k$, and we therefore conclude that under the assumptions of Theorem 5.1, the CDP algorithm run with $k \geq \frac{2 \log(1/\varepsilon) + 3 \log(1/(1-\gamma))}{\log(1/\gamma)}$ also has sample complexity $\widetilde{\Omega}(\varepsilon^{-2}(1-\gamma)^{-3})$.

## F.2 Policy optimisation

Theorem 5.1 can also be straightforwardly adapted to obtain sample complexity results for the *policy optimisation* problem in Markov decision processes, in which the optimal policy must also be learnt from data, prior to estimating its return distribution. We describe one way in which this can be made concrete. Suppose we have a finite-state, finite-action MDP with reward a function of state, with a unique optimal policy $\pi^*$ with an action gap $\varepsilon \in [0, (1-\gamma)^{1/2})$, meaning that at any state $x$, if $a, b \in \mathcal{A}$ are the optimal action and a sub-optimal action respectively, then $Q^{\pi^*}(x, b) < Q^{\pi^*}(x, a) - \varepsilon$. Then, for example, Theorem 1 of Agarwal et al. (2020) yields that for $\delta \in (0, 1)$, with $N = \widetilde{O}(\tilde{\varepsilon}^{-2}(1-\gamma)^{-3} \log(1/\delta))$ sampled transitions per state-action pair we can correctly identify the optimal policy $\pi^*$ with probability at least $1 - \delta$. We may then form the MRP corresponding to executing the policy $\pi^*$, and may resample from the transitions used in identifying the optimal policy to compute approximate return distributions for the optimal policy; Theorem 5.1 and a union bound then guarantee $\varepsilon$-accurate approximations in Wasserstein distance with high probability.

To understand the role of the unique optimal policy assumption in our discussion above, recall that when there are multiple optimal policies, each generally has a distinct collection of return distributions, and so there is not a single object that one should hope to approximate. This is not the case with value functions, which are necessarily identical for all optimal policies. As described in earlier work on policy optimisation with distributional RL (Bellemare et al., 2017, 2023), this lack of uniqueness of optimal return distributions also leads to interesting theoretical issues when analysing algorithms such as value iteration. However, we expect a result in which one obtains an approximation to the return distributions of *some* optimal policy can also be obtained, even in the non-unique-optimal-policy setting.

## F.3 Stochastic rewards

In our main results, we have assumed that the immediate reward is a deterministic function of state. Results in mean-return RL are also commonly given under this assumption (Azar et al., 2013), since this simplifies notation and the extension to stochastic rewards is often straightforward to obtain (Pananjady and Wainwright, 2020). In this section, we describe how to modify the argument presented in the deterministic-reward case to allow for stochastic rewards.

**Assumptions.** We now consider working instead with a reward distribution function $\mathcal{R} : \mathcal{X} \to \mathscr{P}([0, 1])$, so that $\mathcal{R}(x)$ is the *distribution* of immediate rewards received at state $x$. The $N$ i.i.d. transitions obtained from state $x$ are given by $(x, R_i^x, X_i^x)_{i=1}^N$, where as before $X_i^x \overset{\text{i.i.d.}}{\sim} P(\cdot|x)$, and independently $R_i^x \overset{\text{i.i.d.}}{\sim} \mathcal{R}(x)$, for $i = 1, \dots, m$.

**Algorithms.** The categorical operator $T_P$ is defined in the stochastic reward case in direct analogy with the definition presented in Section 4 in the deterministic reward case. The generalisation is that

the quantities $h_{ij}^x$ given in Equation (6), which represent the contribution to mass allocated to $z_i$ at $x$ due to backing up from atom $z_j$, are now defined as

$$h_{ij}^x = \mathbb{E}_{R \sim \mathcal{R}(x)}[h_i(R + \gamma z_j)],$$

for each $x \in \mathcal{X}$ and $i, j = 1, \ldots, m$. The quantities $H_{i,j}^x$ and $T_P$ are defined exactly as in the deterministic case, from this generalised definition of the $h_{i,j}^x$. The model-based algorithms, based on the $N$ sampled transitions at each state described above, now also approximate the $h_{i,j}^x$ from the empirical reward distributions $\hat{\mathcal{R}}(x) = N^{-1} \sum_{i=1}^N \delta_{R_i^x}$, as well as $P$. In particular, we have

$$\hat{h}_{i,j}^x = \frac{1}{N} \sum_{i=1}^N h_i(R_i^x + \gamma z_j),$$

from which we define the corresponding estimators $\hat{H}_{i,j}^x$ of $H_{i,j}^x$. We now write the empirical categorical operator as $T_{\hat{P}, \hat{\mathcal{R}}}$, to emphasise the dependence on the empirical reward distributions $(\mathcal{R}(x) : x \in \mathcal{X})$ too, which is then defined as

$$T_{\hat{P}, \hat{\mathcal{R}}}(x, i; y, j) = \hat{P}(y|x)(\hat{H}_{i,j}^x - \hat{H}_{i,j+1}^x).$$

Model-based DCFP in the stochastic reward setting then simply corresponds to solving the linear system built from $T_{\hat{P}, \hat{\mathcal{R}}}$ appearing in Equation (11), based on this empirical categorical operator.

**Sample complexity.** The model-based DCFP algorithm also satisfies the sample complexity bound given in Theorem 5.3 (and hence in Theorem 5.1 too) in the case of stochastic rewards. Only a few additional details are required to extend the proof in the deterministic reward case to the stochastic case, which we sketch below. As before, we write $F^*$ for the CDF values of the true categorical fixed point and $\hat{F}$ for the estimated fixed point obtained by running DCFP with the operator $T_{\hat{P}, \hat{\mathcal{R}}}$. We also introduce the notation $T_{\hat{P}, \mathcal{R}}$ for the categorical operator using estimated transition probabilities $\hat{P}$ but *true* reward distributions $(\mathcal{R}(x) : x \in \mathcal{X})$, and write $\acute{F}$ for the corresponding categorical fixed point.

**Proof modifications.** First, when performing the initial reduction to categorical fixed-point error, we invoke the triangle inequality an additional time, to obtain

$$\bar{\ell}_2(\eta^*, \hat{F}) \leq \bar{\ell}_2(\eta^*, F^*) + \bar{\ell}_2(F^*, \acute{F}) + \bar{\ell}_2(\acute{F}, \hat{F}). \tag{34}$$

*First term.* As in the proof in the deterministic case, the first term on the right-hand side is $O(\varepsilon)$ due to the assumption on $m$ in the theorem statement; in particular, Proposition 2.2 as proven by Rowland et al. (2018) holds in the case of stochastic rewards too.

*Second term.* The second term on the right-hand side of Equation (34) is bounded by following the same argument as given in Appendices E.3–E.8. The only part of the argument in these sections that specifically relies on the immediate reward being deterministic, rather than relying on the contractivity of the operator (which also holds in the stochastic-reward case (Rowland et al., 2018)), is Proposition B.3, which is used in establishing Proposition 5.11 and Corollary 5.12. We state and prove a version of Proposition B.3 that holds under the weaker assumption of stochastic rewards too.

**Proposition F.1.** *In the general case of stochastic immediate rewards described above, we have that for any $F, F' \in \mathscr{F}$,*

$$\|B_x F - B_x F'\|_{\ell_2}^2 \geq \gamma \|F - F'\|_{\ell_2}^2 - \frac{2}{m(1-\gamma)^{1/2}} - \frac{1}{m^2(1-\gamma)^2} - 4.$$

*Proof.* Following the proof of Proposition B.3, we obtain

$$\|B_x F - B_x F'\|_{\ell_2}^2 \geq \ell_2^2(R + \gamma G, R + \gamma G') - \frac{2}{m(1-\gamma)^{1/2}} - \frac{1}{m^2(1-\gamma)^2},$$

where $R \sim \mathcal{R}(X')$, and independently, $G, G'$ are random variables taking values in $\{z_1, \ldots, z_m\}$ with CDF values given by $F, F'$, respectively, and we write $\ell_2^2(R + \gamma G, R + \gamma G')$ as shorthand for the Cramér distance between the two distributions of these random variables, $\mathbb{E}_{R \sim \mathcal{R}(x)}[(b_{R,\gamma})_\# \nu]$, $\mathbb{E}_{R \sim \mathcal{R}(x)}[(b_{R,\gamma})_\# \nu']$, respectively. We then use the characterisation of the squared-Cramér distance

in Equation (23) to obtain, writing $R_1, R_2 \sim \mathcal{R}(x)$, $G_1, G_2 \sim \nu$, $G_1', G_2' \sim \nu'$, all independent of one another,

$$
\begin{aligned}
&\ell_2^2(R + \gamma G, R + \gamma G') \\
&= \mathbb{E}[|R_1 + \gamma G_1 - (R_2 + \gamma G_2')|] - \frac{1}{2}\mathbb{E}[|R_1 + \gamma G_1 - (R_2 + \gamma G_2)|] \\
&\quad - \frac{1}{2}\mathbb{E}[|R_1 + \gamma G_1' - (R_2 + \gamma G_2')|] \\
&\overset{(a)}{\geq} \mathbb{E}[|\gamma G_1 - \gamma G_2'| - 2] - \frac{1}{2}\mathbb{E}[|\gamma G_1 - \gamma G_2| + 2] - \frac{1}{2}\mathbb{E}[|\gamma G_1' - \gamma G_2'| + 2] \\
&= \gamma \ell_2^2(\nu, \nu') - 4 = \gamma \|F - F'\|_{\ell_2}^2 - 4 \,,
\end{aligned}
$$

as required, where (a) follows from boundedness of the rewards, and the triangle inequality. $\qquad\square$

With Proposition F.1, we obtain a weaker version of Corollary 5.12 in the stochastic reward case, yielding $\|(I - \gamma \hat{P})^{-1} \sigma_{\hat{P}}\|_\infty \leq 6(1 - \gamma)^{-1}$, which is sufficient to conclude the claimed bound on the second term as described in Appendices E.7 & E.8, specifically yielding that this second term is $O(\varepsilon)$ with probability at least $1 - \delta$.

*Third term.* Finally, the third term on the right-hand side of Equation (34) is a new term that emerges specifically in the stochastic-reward case, corresponding to errors in the fixed-point solely due to mis-estimated immediate reward distributions. First, by an analogous argument to that in Section E.3, we may write

$$
\acute{F} - \hat{F} = \sum_{k=0}^{\infty} (T_{\hat{P}, \hat{\mathcal{R}}})^k (T_{\hat{P}, \mathcal{R}} - T_{\hat{P}, \hat{\mathcal{R}}}) \acute{F} \,.
$$

We have

$$
\left\| \left[ (T_{\hat{P}, \mathcal{R}} - T_{\hat{P}, \hat{\mathcal{R}}}) \acute{F} \right](x) \right\|_{\ell_2} \overset{(a)}{\leq} \ell_2(\mathcal{R}(x), \hat{\mathcal{R}}(x)) \overset{(b)}{\leq} \sqrt{\frac{\log(2|\mathcal{X}|/\delta)}{2N}}
$$

with probability at least $1 - \delta$. Here, (a) follows by homogeneity of the Cramér distance (Rowland et al., 2018, Proof of Proposition 2), and (b) follows from the DKW inequality (Dvoretzky et al., 1956; Massart, 1990) and a union bound. Then, using contractivity of $T_{\hat{P}, \hat{\mathcal{R}}}$ and the triangle inequality, we obtain

$$
\|\acute{F} - \hat{F}\|_{\ell_2, \infty} \leq \sum_{k=0}^{\infty} \gamma^{k/2} \sqrt{\frac{\log(2|\mathcal{X}|/\delta)}{2N}} = \widetilde{\mathcal{O}}((1 - \gamma)^{-1} N^{-1/2}) \,,
$$

with probability at least $1 - \delta$, which is $O(\varepsilon)$ under the choice of $N$ in the theorem statement. In summary, taking a union over the events corresponding to all concentration bounds used, we obtain that $\bar{\ell}_2(\eta^*, \hat{F})$ is $O(\varepsilon)$ with probability at least $1 - 2\delta$, as required.

# G    Further experimental results and details

In this section, we give full details for the experiment presented in the main paper, and additionally present extended results on several additional environments. All experiments were run using the Python 3 language (Van Rossum and Drake, 2009), and made use of NumPy (Harris et al., 2020), SciPy (Virtanen et al., 2020), Matplotlib (Hunter, 2007), and Seaborn (Waskom, 2021) libraries. As all experiments are tabular, each run uses a single CPU, and timings are reported within the experimental results.

## G.1    Environments

We report results on four MRPs defined as follows:

1. **Chain**: 10 states arranged in a chain $x_1 \leftrightarrow x_2 \leftrightarrow \cdots \leftrightarrow x_{10}$. Each state transitions to its neighbours with equal probability. States 1 and 10 are terminal. Only state 10 has a reward of 1, and all other states have reward 0.

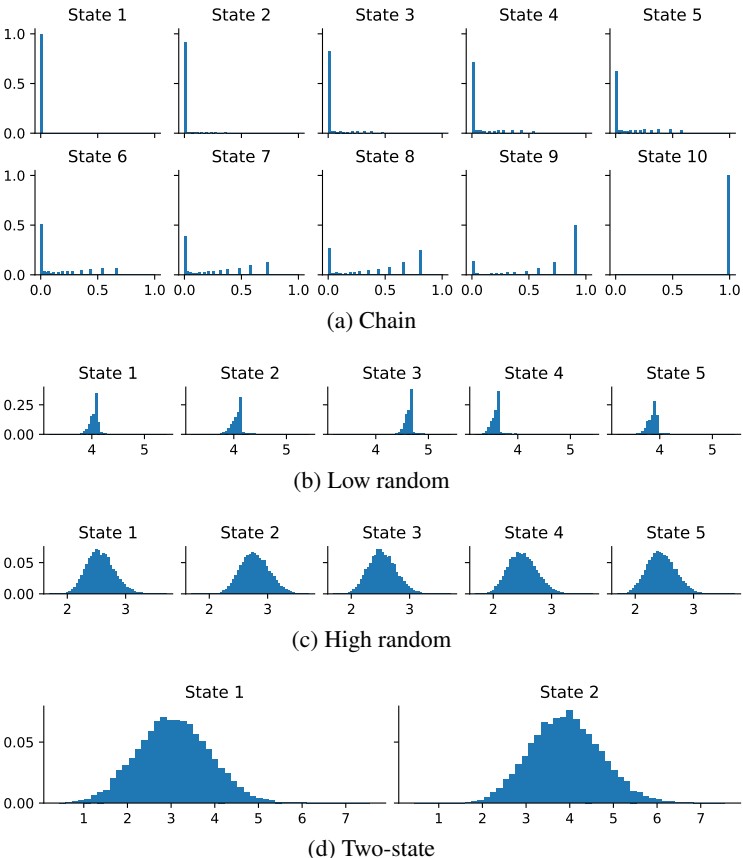

Figure 4: Monte Carlo approximations of return distributions in each of the four environments tested.

2. **Low random**: There are five states, and the transition probabilities from each state to all five states are drawn independently from a Dirichlet distribution with concentration parameter $(0.01, \ldots, 0.01)$. The rewards for each state is drawn i.i.d. from the uniform distribution over $[0, 1]$. We draw these transition probability and rewards using the same random seed to yield the same MRP for all experiments.

3. **High random**: Same as the low random environment, but with transitions drawn from a Dirichlet distribution with concentration parameter $(10, \ldots, 10)$.

4. **Two-state**: A 2-state MRP modified from Example 6.5 of Rowland et al. (2024). The transition matrix is

$$\begin{pmatrix} 0.6 & 0.4 \\ 0.8 & 0.2 \end{pmatrix}$$

where the $(i, j)$ entry is the probability of transitioning from state $i$ to state $j$. State 1 has reward 0, and state 2 has reward 1.

We chose this set of environments to include classic tabular settings such as the chain, two environments with very different levels of stochasticity (low and high random), as well as the two-state example from Rowland et al. (2024), which illustrates the nature of fixed-point approximation error incurred by QDP in environments with tight bootstrapping loops. We vary the discount factor $\gamma \in \{0.8, 0.9, 0.95, 0.99\}$. For reference, we plot Monte Carlo approximations of return distributions for each environment in Figure 4, with $\gamma = 0.9$. The return samples are computed using the first-visit Monte Carlo algorithm. For each initial state, we run at least $T$ transitions such that the maximum error after $T$ transitions $\gamma^T r_{\max}/(1 - \gamma) < \varepsilon$, where $r_{\max} = 1$ and we set $\varepsilon = 10^{-4}$. This is repeated $10^4$ times for each state, giving at least $10^4$ return samples per state.

## G.2 Algorithms

As described in the main paper, we compare DCFP, QDP, and CDP algorithms. For DCFP and CDP, we make use of the sparse structure of the update matrix (see Equation (9)), and implement these algorithms using SciPy's sparse matrix-vector multiplication and linear system solution methods (Virtanen et al., 2020). The mathematical details regarding the sparsity of the update matrix is described in Appendix G.3. For comparison, we also run implementations of DCFP and CDP that do not exploit this sparsity, and instead use standard dense NumPy implementations for matrix-vector multiplication and linear system solution methods; these are denoted by d-DCFP and d-CDP, respectively.

By default the categorical methods (all variants of DCFP and CDP) use the atom locations described in the main paper: $z_i = \frac{i-1}{m-1}(1-\gamma)^{-1}$ for $i = 1, \ldots, m$. These atom locations are sufficient to establish the theoretical results in the main paper, though we note that in practice, particularly when true return distributions are localised within a small sub-interval of $[0, (1-\gamma)^{-1}]$, this setting can be overly conservative. However, in many cases, there are straightforward ways in which the choice of support can be improved, essentially by replacing the uniformly valid return range $[0, (1-\gamma)^{-1}]$ with an *a priori* known environment-specific reward range. As an example, if the known range of immediate rewards forms a sub-interval $[r_{\min}, r_{\max}] \subseteq [0, 1]$, then the return must fall into the environment-specific reward range $[r_{\min}(1-\gamma)^{-1}, r_{\max}(1-\gamma)^{-1}]$, and improved atom locations $z_i = r_{\min}(1-\gamma)^{-1} + \frac{i-1}{m-1}(r_{\max} - r_{\min})(1-\gamma)^{-1}$ (for $i = 1, \ldots, m$) can be used, whilst maintaining the guarantee that the support for the true return distributions lie within this interval. We thus additionally report results for versions of DCFP and CDP that use environment-specific atom locations, to investigate what practical improvements can be gained with such additional information. Specifically, for the high random and low random environments, we use the tighter bounds on support given by known $r_{\min}$ and $r_{\max}$, as described above. In the chain environment, we consider using the advanced knowledge that only the transition into the terminal state is rewarding, yielding a sub-interval for possible returns of $[0, 1]$. In the two-state case, the range of possible returns is the full interval $[0, (1-\gamma)^{-1}]$, and so we do not investigate an environment-specific set of atoms in this case. However, it is clear from the Monte Carlo approximations to the return distributions in this environment in Figure 4 that these distributions do have the vast majority of their probability mass in a much smaller interval than the worst case interval $[0, (1-\gamma)^{-1}]$.

We ran all DP methods with 30,000 iterations. For all categorical DP methods, we verify that the Wasserstein distance between the last iterate to the Monte Carlo ground truth is almost identical to the Wasserstein distance between the categorical fixed-point to the ground truth.

## G.3 Efficient implementation of CDP and DCFP

A straightforward implementation of CDP and DCFP is to vectorise $F$, treating it as a vector indexed by state-index pairs $(x, i)$, and correspondingly treat $T_{\hat{P}}$ as a matrix whose rows and columns are indexed similarly. Iterations of CDP can then be performed by simple matrix-vector multiplications with this representation of $T_{\hat{P}}$, and the solution of the linear system appearing in Equation (11) that forms the core of DCFP can be implemented with a call to a standard linear system solver, such as `numpy.linalg.solve` (Harris et al., 2020).

However, the operator $T_{\hat{P}}$ typically has some specific structure that can be exploited in implementations. In particular, we highlight here the sparse structure of $T_{\hat{P}}$, allowing for potential speed-ups in implementation by making use of sparse linear solvers, such as `scipy.sparse.linalg.spsolve` (Virtanen et al., 2020). Recall that, viewing $T_{\hat{P}}$ as a matrix (c.f. Equation (9)), the element corresponding to the $(x, i)$ row and $(y, j)$ column is given by

$$\hat{P}(y|x)(H_{i,j}^x - H_{i,j+1}^x). \tag{35}$$

In an MRP with sparse transition matrix $P$, the empirical estimate $\hat{P}$ inherits this sparsity, inducing sparsity in $T_{\hat{P}}$. In addition, there is also sparsity induced by the atom index components of the row and column indices, as we now describe. Recall from Equation (8) that $H_{i,j}^x = \sum_{l \leq i} h_l(r(x) + \gamma z_j)$; see Figure 5 for an illustration the function $z \mapsto \sum_{l \leq i} h_l(z)$.

Now, suppose that for some $x \in \mathcal{X}$ and $1 \leq i, j \leq m-1$, $H_{i,j}^x - H_{i,j+1}^x$ is non-zero. This says that the function $z \mapsto \sum_{l \leq i} h_l(z)$ takes on different values at $r(x) + \gamma z_j$ and $r(x) + \gamma z_{j+1}$, and since

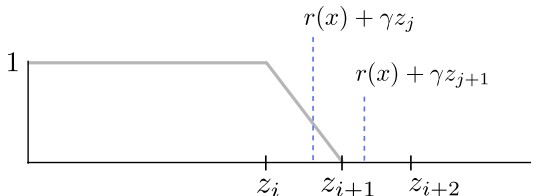

Figure 5: The function $z \mapsto \sum_{l \leq i} h_l(z)$ (grey), and a possible configuration for $r(x) + \gamma z_j$, $r(x) + \gamma z_{j+1}$ in the event of a non-zero $H_{i,j}^x - H_{i,j+1}^x$ term.

the distance between these arguments is $\gamma$ times the grid width $(1-\gamma)^{-1} m^{-1}$, it follows that at least one of these two arguments must lie in the interval $[z_i, z_{i+1}]$; see Figure 5.

From this observation, we deduce two forms of sparsity for the elements given in Equation (35). First, since one of $r(x) + \gamma z_j$ and $r(x) + \gamma z_{j+1}$ must lie in $[z_i, z_{i+1}]$, we deduce that neither of these points can lie in any interval $[z_{i'}, z_{i'+1}]$ with $i' < i - 1$ or $i' > i + 1$, and hence $H_{i',j}^x - H_{i',j}^x = 0$ for such $i'$. From this reasoning, it follows that ranging over $i$ in Equation (35), there are at most 2 non-zero elements. For large $m$, this means that $T_{\hat{P}}$ is very sparse.

Similarly, we can deduce row sparsity by noting that at most $\lceil 2/\gamma \rceil$ indices $j$ can have the property that $r(x) + \gamma z_j$ or $r(x) + \gamma z_{j+1}$ can lie in $[z_i, z_{i+1}]$, and it is only for these indices $j$ that we can have $H_{i,j}^x - H_{i,j+1}^x$ non-zero. So for $\gamma \approx 1$, this also implies sparsity of the elements in Equation (35) as we range over $j$.

### G.4 Results

For each setting, we repeat the experiment 30 times with different sampled transitions. We display trade-off plots (supremum-Wasserstein-1 error and wallclock time) for each of the four environments described above, in Figures 6, 7, 8, and 9, for the cases of $m \in \{30, 100, 300, 1000\}$ atoms and using $N = 10^6$ sample transitions from each state to estimate transition matrix. These curves are obtained by averaging across the 30 repetitions. Some central themes emerge from the results.

In the case when the categorical methods use the environment-independent, standard atom locations $z_i = \frac{i-1}{m-1}(1-\gamma)^{-1}$ for $i = 1, \ldots, m$, we find that for a given atom count $m$, QDP often achieves the lowest asymptotic Wasserstein-1 error. A notable exception is the two-state environment; Rowland et al. (2024) observed that such environments, in which there is a short, high-probability path from a state to itself, can cause high approximation error for QDP, which we believe is the cause of the inaccuracy observed here, particularly at high discounts. However, the categorical approaches often deliver better performance as judged by wallclock time. This is owing to the efficient implementations of the linear operator, and solution methods for the linear system, that are associated with these algorithms. By contrast, the QDP operator is non-linear, and requires a call to a sorting method. We tend to observe the greatest benefits of the sparse implementation for large atom counts, as expected, and also observe the greatest benefits of DCFP over the iterative CDP algorithm in settings with high discount factors. This is also to be expected, since the discount factor controls the rate of convergence of the DP algorithms. Note additionally that the use of environment-specific atom locations generally improves the results obtained with categorical approaches; the improvements in the chain environment are particularly strong, since the narrow return range allows for a denser packing of atom locations, by a factor of $(1-\gamma)^{-1}$, for a given atom count $m$.

We also compare how the supremum-Wasserstein distance changes while the number of samples used to estimate $\hat{P}$ increases from $10^2$ to $10^6$. Since the categorical methods all converge to the same fixed point, we only show the result of DCFP and QDP, using $m \in \{30, 100, 300, 1000\}$ atoms. Figure 10(a) shows that the distance decreases as the number of samples increase. One exception is the QDP with 30 atoms applied to high random at high $\gamma$, where $N = 100$ produced smallest distance on average. Figure 10(b) shows the results when the environment-specific return range is given. This substantially reduced the supremum-Wasserstein distance, especially when using a small number of atoms.

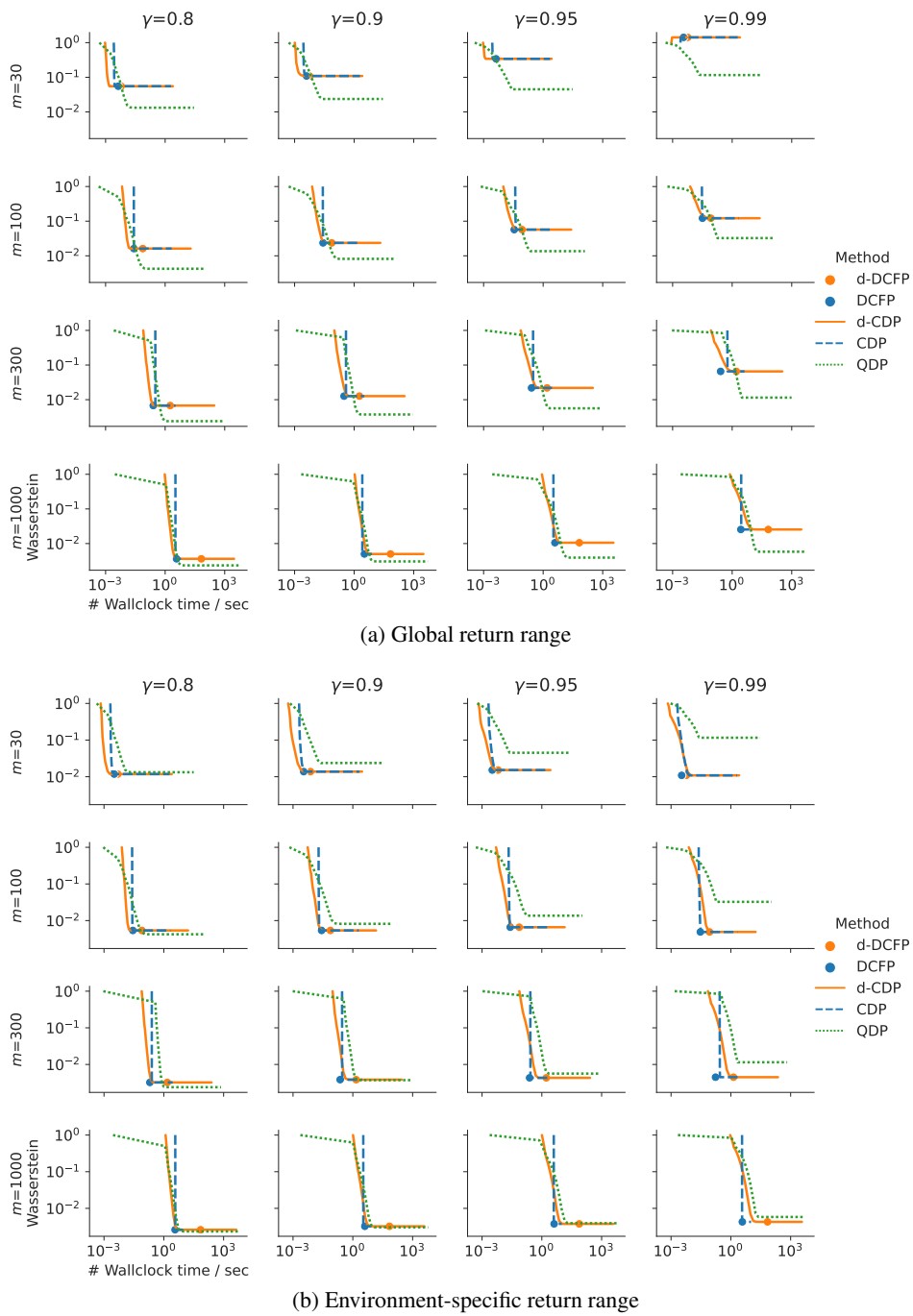

(a) Global return range

(b) Environment-specific return range

Figure 6: Distance vs. run time results for the chain environment, for $\hat{P}$ estimated from $N = 10^6$ sample transitions.

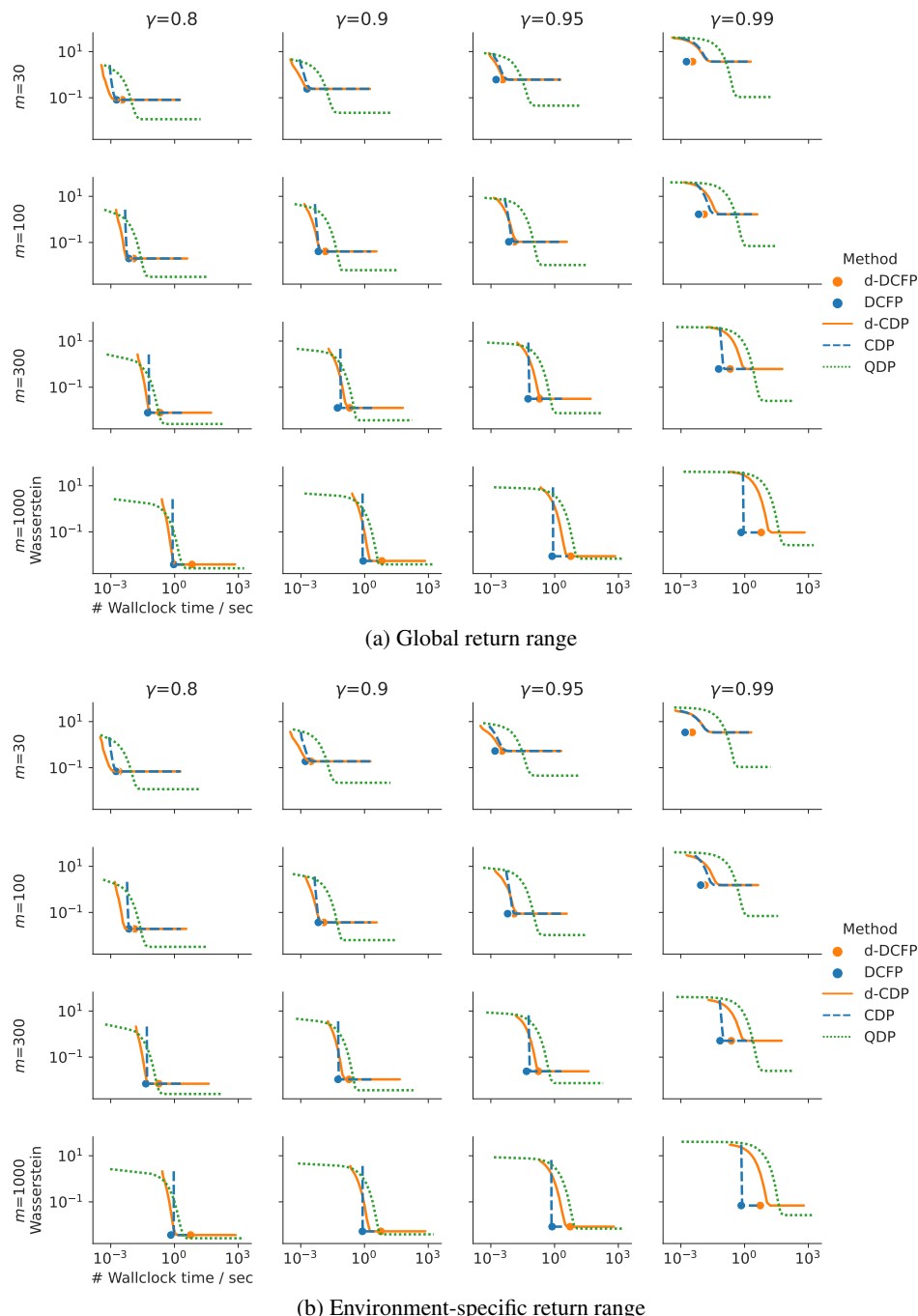

(a) Global return range

(b) Environment-specific return range

Figure 7: Distance vs. run time results for the low random environment, for $\hat{P}$ estimated from $N = 10^6$ sample transitions.

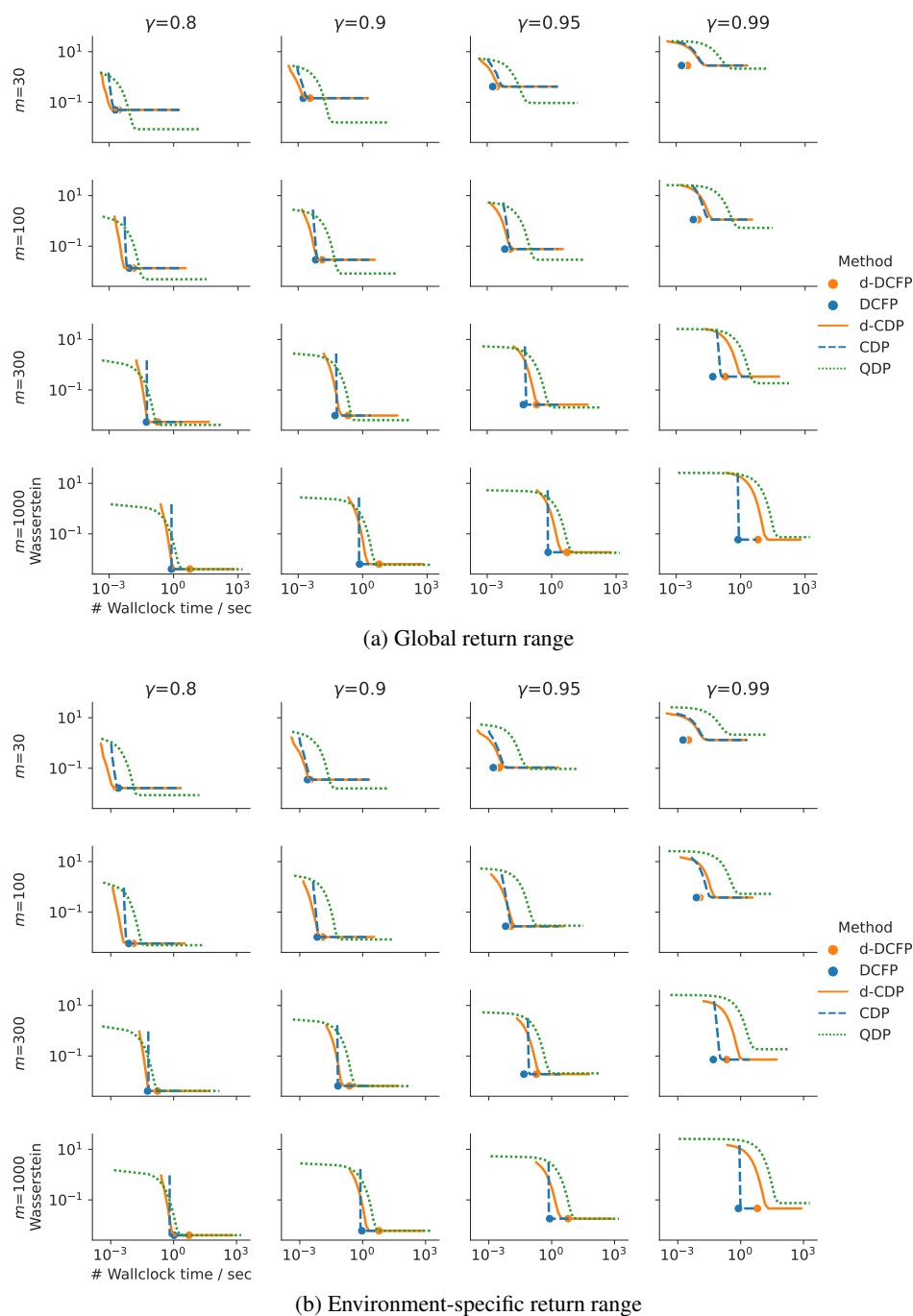

(a) Global return range

(b) Environment-specific return range

Figure 8: Distance vs. run time results for the high random environment, for $\hat{P}$ estimated from $N = 10^6$ sample transitions.

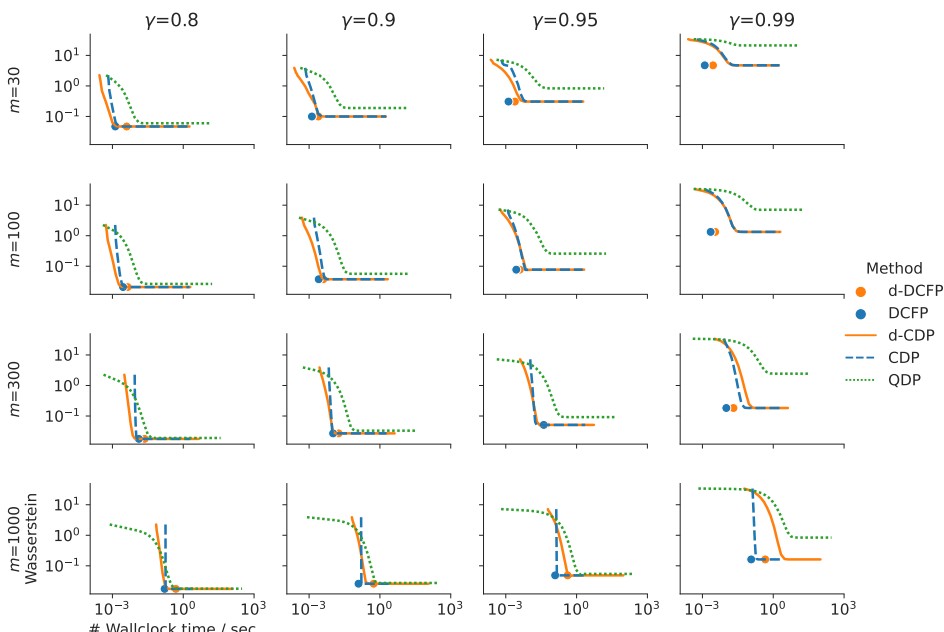

Figure 9: Distance vs. run time results for the two-state environment, for $\hat{P}$ estimated from $N = 10^6$ sample transitions. The return range of this environment coincides with the global return range.

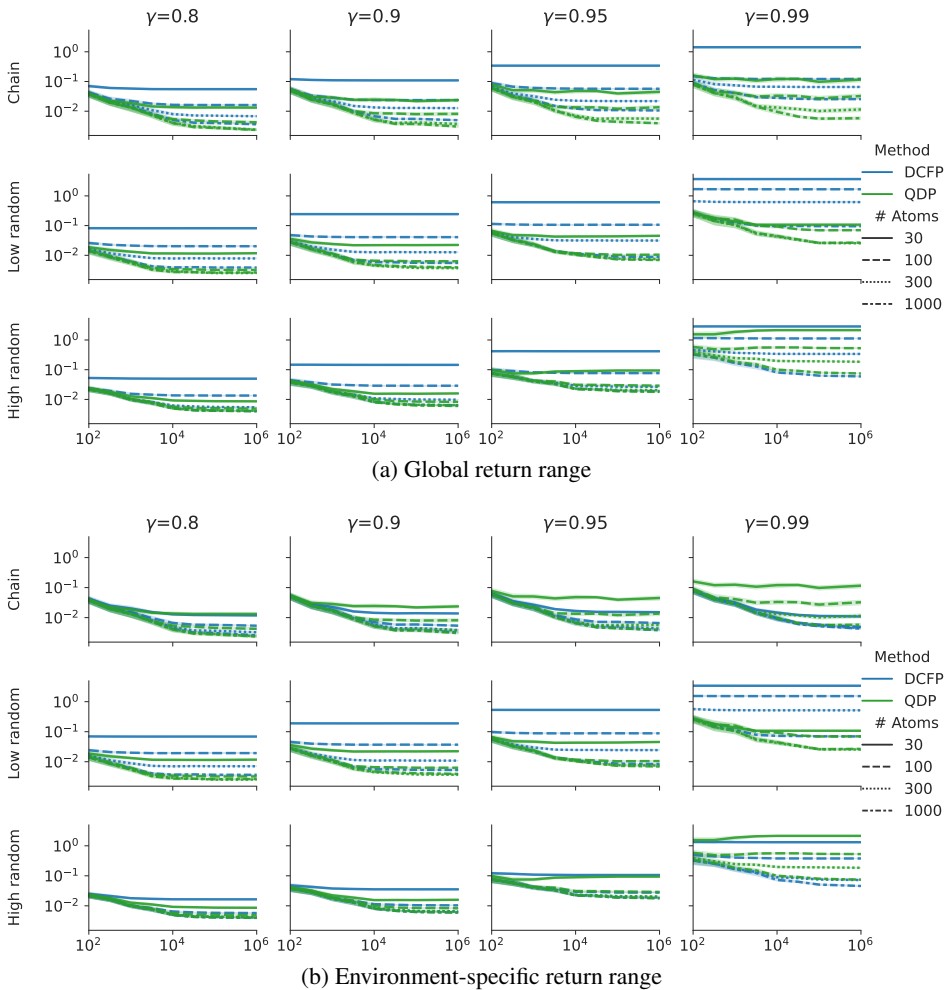

Figure 10: Supremum-Wasserstein distance on convergence. Error envelope indicates 95% confidence interval by bootstrapping. The return range of the two-state environment coincides with the global return range.

## G.5 Example implementations

We provide example implementations for the CDP and DCFP algorithms described in the main paper. We work with updates over cumulative distribution functions, as in Equation (7), matching the form of the operator analysed in the paper. Our intention is to provide a straightforward implementation of these key algorithms, and in particular we have not included optimisations such as exploiting sparsity in the linear solver, as described in Appendix G.3. As shown in our experimental results, there are often significant performance gains that can be obtained from such optimisations, and we encourage practitioners to make use of such optimisations when running DCFP beyond the very smallest instances.

Listing 1 gives an implementation of the categorical Bellman operator itself, Listing 2 gives an implementation of a single CDP update, Listing 3 gives an implementation of the DCFP algorithm in terms of CDFs, and Listing 4 provides a function for converting computed CDF values to corresponding probability mass functions. The code snippets in this paper have the licence below.

```python
# Copyright 2025 Google LLC. SPDX-License-Identifier: Apache-2.0
```

```python
import numpy as np

def hat_functions(values: np.ndarray, support: np.ndarray) -> np.ndarray:
    """Computes the values of the categorical hat functions described in Section 2.3.

    Args:
        values: Values to be projected, of arbitrary shape.
        support: Distribution support to project onto, of shape (n_atoms,).

    Returns:
        Probabilities over support for projected values, of shape (*values.shape, n_atoms).
    """
    support_diff = support[1:] - support[:-1]
    proj_left = (support[1:] - values[..., None]) / support_diff
    proj_left = np.concatenate((proj_left, np.ones_like(values)[..., None]), axis=-1)
    proj_right = (values[..., None] - support[:-1]) / support_diff
    proj_right = np.concatenate((np.ones_like(values)[..., None], proj_right), axis=-1)
    return np.maximum(np.minimum(proj_left, proj_right), 0.)

def construct_cdf_categorical_bellman_operator(
        transition_probs: np.ndarray, rewards: np.ndarray, discount: float, support: np.ndarray
) -> np.ndarray:
    """Constructs the categorical Bellman operator.

    Args:
        transition_probs: Transition probabilities, of shape (n_states, n_states).
        rewards: Reward vector, of shape (n_states,).
        discount: Discount factor.
        support: Vector of atom locations for categorical distributions, of shape (n_atoms,).

    Returns:
        The categorical Bellman operator, as array with shape
          (n_states, n_atoms, n_states, n_atoms).
    """
    # Indices: (x, j)
    bootstrap_returns = rewards[:, None] + discount * support[None, :]
    # Indices: (x, j, i)
    h = hat_functions(bootstrap_returns, support)
    # Indices: (x, i ,j)
    h = np.transpose(h, (0, 2, 1))
    H = np.cumsum(h, axis=1)
    H_diff = -np.diff(H, append=0., axis=2)
    return np.einsum('xy,xij->xiyj', transition_probs, H_diff)
```

Listing 1: Construction of the categorical Bellman operator over CDFs.

```python
def apply_categorical_bellman_operator(operator: np.ndarray, F: np.ndarray) -> np.ndarray:
    return np.einsum('xiyj,yj->xi', operator, F)
```

Listing 2: A single step of CDP implemented by an application of the categorical Bellman operator.

```python
def dcfp(
        transition_probs: np.ndarray, rewards: np.ndarray, discount: float, support: np.ndarray
) -> np.ndarray:
    n_states = transition_probs.shape[0]
    n_atoms = support.shape[0]
    operator = construct_cdf_categorical_bellman_operator(transition_probs, rewards, discount, support)
    # Constructing the linear system in Equation (11)
    H_tilde = np.sum(operator[:, :-1, :, -1], axis=-1)
    H_tilde_flat = H_tilde.flatten()
    T_tilde = operator[:, :-1, :, :-1]
    n_dim = n_states * (n_atoms - 1)
    T_tilde_flat = np.reshape(T_tilde, (n_dim, n_dim))
    # Solving the linear system in Equation (11)
    F_tilde_flat = np.linalg.solve(np.eye(n_dim) - T_tilde_flat, H_tilde_flat)
    F_tilde = np.reshape(F_tilde_flat, (n_states, n_atoms - 1))
    F = np.hstack((F_tilde, np.ones((n_states, 1))))
    return F
```
Listing 3: An example implementation of DCFP.

```python
def cdf_to_pmf(cdf_values: np.ndarray) -> np.ndarray:
    return np.diff(cdf_values, axis=-1, prepend=0.)
```
Listing 4: Conversion of computed CDF values to probability masses.

It is also possible to implement the operator to take probability mass function values as input rather than CDF values, as in Equation (6); this essentially amounts to a change of basis, and can be achieved by using h rather than H_diff in the np.einsum call in the return line of the construct_cdf_categorical_bellman_operator function; see Listing 5. For implementations of CDP, this implementation of the operator may be preferable for its interpretability, more closely aligning with implementations of categorical temporal-difference learning (Bellemare et al., 2017) that are implemented in terms of probability mass functions, such as in the RLax library (DeepMind et al., 2020). It is also possible to implement DCFP with the probability mass function version of the operator. In comparison with the implementation in Listing 3, the normalisation constraints to be added (as discussed in Section 4.1) are no longer constraints on individual variables of the form $F_m(x) = 1$, but rather take on the form $\sum_{i=1}^{m} p_i(x) = 1$.

```python
def construct_pmf_categorical_bellman_operator(
        transition_probs: np.ndarray, rewards: np.ndarray, discount: float, support: np.ndarray
) -> np.ndarray:
    # Indices: (x, j)
    bootstrap_returns = rewards[:, None] + discount * support[None, :]
    # Indices: (x, j, i)
    h = hat_functions(bootstrap_returns, support)
    return np.einsum('xy,xji->xiyj', transition_probs, h)
```
Listing 5: An implementation of the categorical Bellman operator over probability mass functions.

