# OpenReview forum: "Near-Minimax-Optimal Distributional Reinforcement Learning with a Generative Model"
_NeurIPS.cc/2024/Conference — NeurIPS 2024 poster_

### Official Review · Reviewer_WbyR · 2024-07-08

**Soundness:** 3
**Presentation:** 4
**Contribution:** 3
**Rating:** 6
**Confidence:** 3

**Summary:**

This paper proposes a novel model-based algorithm for policy evaluation in distributional RL and shows its near optimality within the generative model framework.  Additionally, they conduct an experimental study comparing their algorithm with quantile DP in a synthetic environment.

**Strengths:**

Discretizing the distribution's support and then rewriting the update as a linear mapping is an important algorithmic contribution of this paper and is particularly interesting. It reveals a connection between categorical algorithms and linear systems---and, as they did, we can reduce the former to the latter. The paper is well-written. The algorithm and its main idea are clearly presented. The theory looks sound.

**Weaknesses:**

Why do you assume N iid samples for *each state* (line 56)? The common wisdom in policy evaluation under generative model is that more samples are needed for states frequently visited by the policy and fewer samples are needed for states that are less visited. As an extreme case, consider a state that is entirely unreachable---in this case no samples would be needed for that state, and the theoretical analysis should still hold. Thus, I wonder if the results in this paper can be achieved under that assumption.

According to the experimental results, the proposed algorithm does not outperform quantile DP in terms of sample efficiency. However, it has significantly better running time.

**Questions:**

Is the cubic dependence on the state space for computational efficiency good?

**Limitations:**

The authors discussed the limitations, and I found no potential negative societal impact.

---

> ### Author Rebuttal · Authors · 2024-08-06
>
> Thank you for your summary of our work, your feedback and your questions. We are glad to hear you found the paper well-written, the algorithm and main idea clear, and the theory sound.
>
> We are glad also that you found the connection between categorical DP and linear mappings interesting. This is one of the key observations underpinning our theoretical results in this paper, and we expect this observation alone will be important for both implementation and further analysis of categorical-based algorithms in the future.
>
> $N$  **iid samples at each state.**
>
> * The task we focus on is to estimate return distributions conditioned on each starting state for the environment (this is the typical case in the literature on estimation with generative models: see for example Azar et al., 2013; Pananjady and Wainwright, 2020), so there is no notion of an unreachable state in this problem.
>  * We also note that in the standard problem set up, we cannot know in advance whether any particular state is unreachable from any other; we would have to infer this from samples we observe.
>  * The reason that $N$ iid samples are used at each state is that this is sufficient to obtain a sample complexity bound that essentially matches the lower bound; this observation goes back to Azar et al. (2013) in the case of expected returns. This shows that more complex strategies for determining the number of samples to be taken at each state (for example, using adaptive strategies based on the estimated level of stochasticity in transition distributions based on samples seen so far) cannot improve the functional form of the sample complexity.
>
> However, we believe what the reviewer suggests may be an interesting direction for further research questions, such as:
>  * For the distinct problem where some information about the environment is revealed in advance, before any samples are observed (e.g. states that are reachable with low, or zero, probability, from some fixed initial state), should the sampling strategy be adapted?
>  * Even when no information is revealed in advance, can we adapt our sampling process on the basis of the samples observed? For example, if a particular state has never been transitioned to in our dataset, and we are not interested in estimating the return distribution conditioned on this state as the starting state, can we safely avoid sampling transitions from this state? As described above, such approaches cannot improve the functional form of the sample complexity, but may lead to empirical improvements in some settings.
> Answering this question would likely constitute a full new paper, requiring substantial analysis of adaptive confidence bounds for simplex-valued random variables and their interaction with the algorithms described in our paper.
>
> **Cubic dependence on state space.**
>  * This cubic dependence ultimately stems from the fact that even in the non-distributional case, practical solution of the linear system $V^\pi = r^\pi + \gamma P^\pi V^\pi$ has cubic complexity in the state space size.
>  * The focus of our paper is on sample complexity (in particular, in the sense described in the paper, showing that learning return distributions is not more statistically complex than learning mean returns, when using our categorical algorithm). However, we do describe possible computational improvements at various points in the paper, as we expected this to be a topic of interest to a variety of readers. See for example:
>    * Appendix G.3 for discussion of efficient implementation considerations, such as exploiting sparsity in the operators concerned.
>    * Appendix F.1 for discussion of categorical dynamic programming approaches which do not compute the exact fixed point, but for which theoretical guarantees are still obtainable (referred to just prior to Section 5.1 in the main text).

---

> > ### Comment · Reviewer_WbyR · 2024-08-11
> >
> > Thanks for the detailed feedback. I will keep my positive score.

---

### Official Review · Reviewer_Wuoe · 2024-07-09

**Soundness:** 3
**Presentation:** 3
**Contribution:** 2
**Rating:** 5
**Confidence:** 4

**Summary:**

The paper presents a novel algorithm for model-based distributional RL and establishes that it achieves near-minimax-optimal performance for approximating return distributions in the generative model regime. This is a significant contribution, showing that the distributional RL is sample-efficient with a generative model, and being the first work providing practical implementation guidance to achieve this.

**Strengths:**

The paper introduces the new algorithm DCFP for distributional RL, which directly approximates the cumulative probability.  The development of the DCFP algorithm is a key contribution. The DCFP algorithm effectively leverages a generative model to achieve efficient performance.

The authors show that DCFP achieves minimax lower bounds for sample complexity in Wasserstein distance, thus addressing a question raised by Zhang et al. (2023).  Besides, the paper's theoretical contributions, such as the introduction of the stochastic categorical CDF Bellman equation and the detailed derivations of the minimax-optimal sample complexity, are significant but may be challenging for readers without a strong background. I believe this work provides a deeper understanding of the field of distributional RL.

**Weaknesses:**

I have some concerns about the experiments. The authors claim that DCFP is generally faster than QDP. I feel this conclusion is primarily due to computationally costly sort operations in QDP. I am also curious about whether both DCFP and QDP are generally lower than CDP, as DCFP also involves costly matrix inverse operations. It would be better to show some results of CDP in the main paper. Additionally, the authors mentioned  $T_P$ has some sparse structure allowing for potential speed-ups and making use of sparse linear solvers from the scipy package in implementation. If solving this linear system using these solvers is stable and whether the results remain reliable as the state space scales up.  It would be helpful to discuss these.

**Questions:**

Q1: The algorithm operates under the generative model assumption, where it can sample from any state-action pair. Since I am not particularly familiar with this field, could you elaborate on the justifiability of this assumption?  In my limited knowledge, this assumption seems quite far away from reality, and discussing its limitations and applicability would be beneficial.

Q2 The proposed DCFP algorithm aims to align theoretical analysis with practical implementation and can be seen as a modified category-based distributional RL algorithm. However, However, is the sacrifice in computational efficiency deserved?  It would be better to discuss the scalability of the proposed DCFP algorithm in real-world, large-scale environments.

Q3: The current algorithm is model-based and solving this algorithm relies on constructing the empirical transition probabilities. Does this suggest that DCFP is not easily applicable in the model-free setting? In other words,  does the idea of directly learning the cumulative probability mass rather than discrete probability mass face additional challenges in a model-free setting?  Furthermore, if sample-efficient distributional RL can only be achieved under a generative model assumption?

**Limitations:**

Please refer to the question and weakness part.

---

> ### Author Rebuttal · Authors · 2024-08-06
>
> Thank you for your feedback and positive assessment of our work. We are glad to hear you believe this work provides a deeper understanding of distributional RL.
>
> **Experiments.**
>  * *Comparison with QDP.* The reviewer is correct that sorting in the QDP algorithm contributes to its higher computational complexity (and higher wall-clock time in experiments). One of the takeaways from the paper is to highlight the nice linear structure of categorical updates in contrast to the structure of the QDP update. We discuss this in Section G.4, and would be happy to add further discussion if there are any further points the reviewer would like to see highlighted.
>  * *CDP results.* Our paper is primarily theoretical (focusing on sample complexity, and key tools for better understanding categorical distributional RL), and the main paper experiments are intended to provide a brief illustration of implementations of the algorithms described in the main paper. However, to complement the main theoretical contributions of the paper, we do include detailed comparisons against CDP in the appendix.
>  * *Sparse structure and implementation details.* We discuss these aspects in more detail in Section G.3, and speaking generally have found the implementations to be very stable in practice, as they rely on straightforward linear algebra. Please let us know if you have any further questions on this aspect.
>
> **Question 1.**
>  * The generative model setting is a standard assumption in statistical analysis of reinforcement learning algorithms. Some references included in the paper that study RL algorithms under this assumption include (Kearns et al., 2002; Kakade, 2003; Azar et al., 2013; Sidford et al., 2018; Agarwal et al., 2020; Pananjady & Wainwright, 2020; Li et al., 2020).
>  * For further background, the generative model setting can be thought of as removing the exploration aspect of the RL problem, and allows a clear framing of the purely statistical question as to how many samples a given algorithm needs to compute accurate estimators. This is the approach taken in many analyses of RL algorithms (see references above), and insights in the generative model setting often reveal techniques in tackling other settings too. In our case, our analysis contributes key insights (such as the linear structure of the categorical operator, and the stochastic categorical CDF Bellman equation) which will be useful in improving analyses and algorithms in distributional RL. In fact, we have already seen these ideas being picked up and applied in analyzing temporal-difference learning distributional RL algorithms.
>
> We would be happy to expand the discussion in the final version of the paper, please let us know if there are any further questions you have on this topic.
>
> **Question 2.**
>
> DCFP has cubic complexity in the state space, in common with approaches in non-distributional RL that aim to directly compute value functions by solving linear systems of equations. However, we emphasize that our general analysis approach is also applicable to standard categorical DP (with a sufficient number of updates), and so a sacrifice in computational efficiency is not necessarily required. We discuss extensions of the analysis to CDP in Appendix F.1, and provide an empirical comparison in Appendix G. In general, the relative computational merits of DCFP and CDP will depend on several factors such as state space size, discount factor, and any exploitable structure in the operator such as sparsity. Our further experimental results in Appendix G give a sense of these trade-offs.
>
> **Question 3.**
>  * DCFP is intrinsically model-based. However, this is orthogonal to the idea of learning cumulative probabilities and discrete probabilities, which can be translated between as described in Section 4.1. We find the cumulative probability formulation neater in obtaining a linear system with a unique solution, which is why we settle on this presentation in Section 4.1. We hope this clears up the comment contrasting cumulative mass and discrete mass, but let us know if anything remains unclear.
>  * To the best of our knowledge, there is indeed a gap in the sample complexity obtainable under the generative model assumption, and under e.g. online interaction. However, we emphasize that the technical tools we contribute here (linear structure of categorical operator, stochastic CDF Bellman equation) are already useful in analyzing distributional RL algorithms under other modeling assumptions, such as online interaction.

---

> > ### Author Response · Authors · 2024-08-12
> >
> > Thank you again for your review of our work. As the discussion period is coming to an end, we wanted to check whether you have any further queries after reading our rebuttal?

---

> > > ### Comment · Reviewer_Wuoe · 2024-08-13
> > >
> > > Thanks for your response. I do not have further questions.

---

### Official Review · Reviewer_HAEX · 2024-07-10

**Soundness:** 3
**Presentation:** 3
**Contribution:** 3
**Rating:** 7
**Confidence:** 4

**Summary:**

The authors propose a new algorithm for distributional reinforcement learning under the generative model setting with categorical representation. New upper bound for sample complexity is presented for the proposed algorithm. Some empirical results comparing the method with other alternatives are presented, showcasing its benefits over the alternatives.

**Strengths:**

The main algorithm as well as the sample complexity results are solid contribution to the reinforcement learning community. Although I did not check the proofs, the results seem believable.

**Weaknesses:**

The generative model setting is rather limited. The results do not directly contribute towards learning a better policy, but I assume some of the tools presented will find their uses in many scenarios.

**Questions:**

How about the all-gaussian setting where every reward distribution is a gaussian?

---

> ### Author Rebuttal · Authors · 2024-08-06
>
> Thank you for your feedback and positive assessment of our work.
>
> On the significance of the work, we believe many of the tools we have developed in the paper (such as the interpretation of the categorical operator as a particular linear map, the stochastic categorical CDF Bellman equation, the availability of a closed-form solution to the categorical Bellman equation) will play a key role in implementations and analysis of categorical distributional RL in general. We have already seen our work used in this way to develop analyses for temporal-difference algorithms for distributional RL based on categorical parameterisations.
>
> **Gaussian setting.** This is a great question. The core algorithms described in the paper can be applied in this setting too. The algorithms, and the ways in which the analysis can be applied in this more general setting, are described in Section F.3 (referred to just before Section 5.1 in the main text). The central idea is that the variables $h^x_{ij}$ are now defined as expectations under these reward distributions (see Line 1054), which can be accurately approximated via standard numerical integration libraries such as scipy.integrate; this applies to many different classes of distributions, not only Gaussians.

---

> > ### Author Response · Authors · 2024-08-12
> >
> > Thank you again for your review of our work. As the discussion period is coming to an end, we wanted to check whether you have any further queries after reading our rebuttal?

---

> > ### Comment · Reviewer_HAEX · 2024-08-13
> >
> > Thanks for the response, I'll keep my score.

---

### Official Review · Reviewer_1p2t · 2024-07-12

**Soundness:** 2
**Presentation:** 2
**Contribution:** 2
**Rating:** 5
**Confidence:** 3

**Summary:**

This paper proposes a min-max optimal algorithm for model-based distributional RL, which is used to approximate the return distributions under the assumption of having access to generative model. New theoretical analysis is provided for categorical approaches in distributional RL, with the introduction of a new distributional Bellman equation and the stochastic categorical CDF Bellman equation. Empirical studies are performed for benchmark comparison with other model-based distributional RL algorithms.

**Strengths:**

The main contribution of the paper is the theoretical results for distributional RL by assuming the access of a generative model, where the goal is to estimate the full distribution of returns at each state instead of just the expected returns. In particular, this work exhibits several promising results:

1. Authors establish theoretical guarantees for the proposed distributional RL algorithm, i.e. the direct categorical fixed-point algorithm (DCFP), which directly computes the fixed point of CDP (an existing distributional RL algorithm). It achieves a sample complexity that matches with the lower bound up to logarithmic factors when estimating return distribution in Wasserstein distance.

2. Theoretically, authors introduces a new distributional Bellman equation and the stochastic categorical CDF Bellman equation for categorical distributional RL.

**Weaknesses:**

While this paper provides insights into the distributional RL, there are still potential improvements that authors can further consider:

1. It is unclear about the motivation of considering categorical approaches to distributional RL and how the proposed algorithm excels existing distributional RL approaches throughout the text. While it makes sense to choose a tractable representation of probability distributions for approximation, the representation considered as in line 81 - 88 appears to be simple and may have limited expressiveness in practice.

2. The convergence result provided in Proposition 2 does not categorize the convergence rate, which can be essential for computational efficiency.

3. While the theoretical results are interesting, the practical applicability is not quite clear. Numerical results are presented on a 5-state simple tabular environment. It will be beneficial to include a practical example to better illustrate how the proposed algorithm can be particularly useful.

**Questions:**

1. In the studied setting, how does "policy" and "actions" come into play when evaluating return / value functions? Do you consider fixed policy?

2. In Section 2, it seems that the randomness being considered comes from transition, reward function and initial state distribution. How about the randomness in policy?

**Limitations:**

This is a theoretical work, no potential negative social impact

---

> ### Author Rebuttal · Authors · 2024-08-06
>
> Thank you for your review and feedback on our work. We believe we address all concerns raised in the review, and would welcome any further questions.
>
> **Weakness 1: Categorical approximations.** These approximations are well established in distributional reinforcement learning. They have been applied successfully in a variety of practical settings, such as in deep reinforcement learning in simulation (Bellemare et al., 2017), environments based on clinical decision-making (Böck & Heirzinger, 2022), and robotics (Haarnoja et al., 2024). Further, Proposition 2.2 (due to Rowland et al., 2018), and specifically the bound in Equation (5), establishes theoretically that these parametrizations can learn arbitrarily accurate approximations to the true return distributions, with error decaying as $O(1/\sqrt{m})$ as measured by Cramér distance, so there is both empirical and theoretical support that the categorical approximations used are sufficiently rich.
>
> **Weakness 2: Convergence result.** We don't know what the reviewer is referring to by "Proposition 2". If Proposition 2.2 is meant, then please see the comment above regarding the fact that this bound *does* characterize the convergence rate of the approximate distribution to the true return distribution as a function of m. Note that Theorem 5.1, our main theoretical result, also characterizes the convergence rate as a function of number of samples.
>
> **Weakness 3: Applicability.** We have included a small suite of experiments on environments in the appendix, varying qualities of the environment such as levels of stochasticity, to exhibit the method in a range of settings. Having said this, this is essentially a theoretical paper, and our core contribution is to show the (perhaps surprising) result that, in the sense described in the paper, no more samples are needed to estimate full return distributions accurately compared to just their means. We expect many of the tools developed for our analyses (such as the linear structure of the categorical Bellman operator, the stochastic CDF Bellman equation, the availability of a closed-form solution to the categorical Bellman equation) will be useful to the RL community in general in developing implementations and analyses of categorical-based algorithms, and we have already seen our techniques applied in analyzing temporal-difference learning versions of categorical algorithms.
>
> **Question 1: Actions.** The reviewer is right, we consider a fixed policy, and this allows us to consider a Markov reward process (rather than a Markov decision process) as described in Section 2, and avoid including actions in our notation, which leads to more concise expressions throughout the paper. This is a common approach when describing algorithms for evaluation of a fixed policy in reinforcement learning, see for example Pananjady & Wainwright (2020) for a recent example, and Section 2.3 of Szepesvári (2010) for discussion in the context of a textbook.
>
> In more detail, given an MDP with transition probabilities $p : \mathcal{X} \times \mathcal{A} \rightarrow \mathscr{P}(\mathcal{X})$ and a fixed policy $\pi : \mathcal{X} \rightarrow \mathscr{P}(\mathcal{A})$, we can define a corresponding Markov reward process where the transition probabilities $P : \mathcal{X} \rightarrow \mathscr{P}(\mathcal{X})$ are defined by $P(x'|x) = \sum_a \pi(a|x) p(x'|x,a)$, so that the randomness owing to action selection is "folded into" the transition probabilities of the MRP. Reward distributions for the MRP are constructed similarly.
>
> We would be happy to add discussion along these lines to the final version of the paper, please let us know if you have any further questions on this point.
>
> **Question 2: Randomness.** As described in the answer to the point above, the randomness in the policy is folded into randomness in the next-state and reward distributions in defining the Markov reward process.
>
> **Additional references**
>
> Haaronja et al. (2024). *Learning Agile Soccer Skills for a Bipedal Robot with Deep Reinforcement Learning.*
>
> Szepesvári (2010). *Algorithms for Reinforcement Learning.*

---

> > ### Comment · Reviewer_1p2t · 2024-08-12
> >
> > I thank the authors their response.
> >
> > It sounds to me the formulation of Markov reward process will be able to easily borrow the tools from Markov chain analyses, which will be indeed beneficial and interesting.
> >
> > I am curious if generalizing to the case of random policies for policy evaluation in distributional RL (where the randomness cannot be folded into the described process), what will be the main technical challenges in analysis? Whether partial results of the current analysis can be utilized or adapted to handle this scenario?

---

> > > ### Author Response · Authors · 2024-08-12
> > >
> > > Thank you very much for the further questions. We have provided answers below, please let us know if you have further queries on these or other topics.
> > >
> > > To be clear, our analysis does handle the case of a random policy in an MDP, and this is one of the main motivating cases for our analysis. To mathematically define the corresponding Markov reward process, the transition probabilities are defined as in our comment above, and the reward distributions are defined similarly. It may be that this leads to stochastic rewards in the MRP, rather than the deterministic reward function we assume in the main paper. It's a great question from the reviewer as to how the analysis adapts to this setting, and we describe this in detail in Appendix F.3. Essentially, the core techniques for the proof are exactly the same, and just an additional step is required in the argument to ensure that we have accurate estimates of immediate reward distributions, with no change to the sample complexity rate.
> > >
> > > You might also be interested in the setting where we want to evaluate the unknown optimal policy in an MDP; this can also be handled by our analysis. We describe this extension in Appendix F.2, in which first an algorithm is run to identify the optimal policy with high probability, and then our evaluation procedure produces an estimate of the return distributions of this policy.

---

> > > > ### Comment · Reviewer_1p2t · 2024-08-12
> > > >
> > > > Appreciate your further clarification.
> > > >
> > > > At this point, I do not have further questions and keep my score unchanged. Look forward to the revision!

---

### Decision · Program_Chairs · 2024-09-25

**Decision:**

Accept (poster)

**Comment:**

This paper proposes a min-max optimal algorithm for model-based distributional RL, which is used to approximate the return distributions under the assumption of having access to generative model. New theoretical analysis is provided for categorical approaches in distributional RL, with the introduction of a new distributional Bellman equation and the stochastic categorical CDF Bellman equation. Empirical studies are performed for benchmark comparison with other model-based distributional RL algorithms.

All reviewers believe this paper is above the bar of acceptance. The AC agrees and recommends acceptance.